# HOW DO DIFFUSION MODELS LEARN AND GENERALIZE ON ABSTRACT RULES FOR REASONING?

## ABSTRACT

Diffusion models excel in generating and completing patterns in images. But how good is their ability to learn hidden rules from samples and to generate and reason according to such rules or even generalize to similar rules? We trained a wide family of unconditional diffusion models on Raven's progression matrix task to precisely study this. We quantified their capability to generate structurally consistent samples and complete missing parts according to hidden rules. We found diffusion models can synthesize novel samples consistent with rules without memorizing the training set, much better than GPT2 trained on the same data. They memorized and recombined local parts of the training samples to create new rule-conforming samples. When tasked to complete the missing panel with inpainting techniques, advanced sampling techniques were needed to perform well. Further, their pattern completion capability can generalize to rules unseen during training. Further, through generative training on rule data, a robust rule representation rapidly emerged in the diffusion model, which could linearly classify rules at 99.8% test accuracy. Our results suggest diffusion training is a useful paradigm for reasoning and learning representations for downstream tasks even for abstract rules data.

## 1 INTRODUCTION

> "When I have presented one corner of a subject, and they cannot tell me the other three corners from it, I do not repeat my lesson." – Confucius, *The Analects*

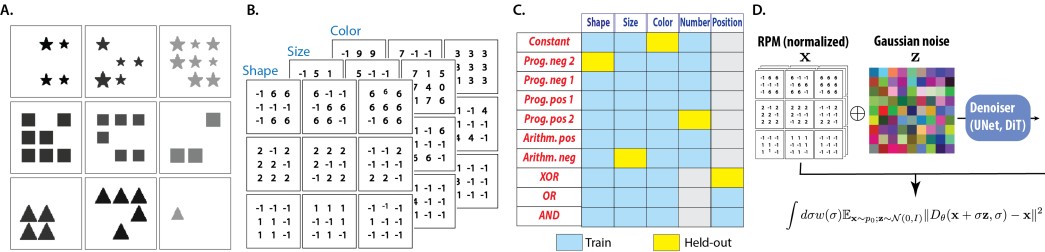

Figure 1: **Design of the study A.** Example Raven's progression matrix, **B.** and its encoding as 3x9x9 integer array. The underlying rule is constant shape. **C.** The 40 relational rules, with 5 rules held out during training. **D.** Training of diffusion models on RPM data.

Human excels at discovering regular structure from a small number of samples, and they can apply such rule to novel setting to generate new samples or completing missing part based on the same rule. The Raven's progressive matrix (RPM) (Raven, 1936) is a famous task in human reasoning literature. In the generative version of this task (GenRAVEN), the subject observes two complete rows of panels and is tasked to complete the third row in a manner that is consistent with the first two rows (Fig.1A). Ideally, the subject needs to infer the underlying rule consistent with the first two rows and applies it correctly to the third row. How can we train a learning system to solve such reasoning task?

If we conceptualize all rule-conforming samples as a joint distribution, then rule learning can be framed as a generative modelling problem or learning the correct joint distribution. Further, reasoning about the missing panel can be framed as sampling from the conditional probability (Prystawski et al.,

2024). One conceptual problem is, given finite training samples, the rule governing them, or the 'true' joint distribution is under-specified: in one extreme, the joint is the empirical distribution of the training set; in the other extreme, the joint distribution can be overly broad and might not align with what we mean by rule. Fundamentally, the rule and the distribution learned by the system should be affected by its inductive bias, be it human or AI. Given this ambiguity, we asked whether modern generative AI systems could learn the correct "joint distribution" given finite samples. If so, can they reason about the missing part through conditional sampling?

So in this work, we focus on precisely studying the "reasoning" capability of diffusion models using a popular reasoning tasks, the Raven's Progressive Matrices (RPM). We trained diverse kinds of diffusion models (EDM, DiT, SiT) on the RPM data and systematically evaluated their performance in generating samples, classifying samples and completing missing panels consistent with underlying rules. We focus on answering the following questions: Can diffusion models learn abstract rules from training data? If so, on what level are they memorizing and on what level are they innovating? What are the factors affecting rule learning and what are their limitations? Note that, we are not only motivated to find the best model that learns rules, but also to highlight the universal aspects of the structural learning process in diffusion models.

**Main Contributions**    The main contributions of our work are the following,

- Demonstrating diffusion models can learn to generate novel samples following abstract rules, with minimal memorization of the training set, esp. compared to autoregressive models.
- Showing diffusion models learn to create new samples according to rules by recombining local parts memorized in training.
- Characterizing the hierarchical learning dynamics of diffusion models when learning data with hierarchical rule structure.
- Showing a robust representation that perfectly classifies rules emerged in diffusion models through generative training.
- Showing unconditional diffusion models can be used for conditional inference, i.e. completing missing panels consistent with the rule.
- Illustrating the data scaling of rule learning: diffusion model starts to learn abstract rule structure when it fails to memorize (parts of) the sample.

## 2 RELATED WORK

**Solving RAVEN's task with learning method**    Developing models for abstract reasoning is a central goal in machine learning, with the RPM task widely used as a benchmark. Large-scale datasets like PGM (Barrett et al., 2018) and RAVEN (Zhang et al., 2019a) have facilitated the development of deep learning models, such as those based on CNNs (Zhang et al., 2019b; Spratley et al., 2020; Zhuo & Kankanhalli, 2022), relation networks (Barrett et al., 2018; Benny et al., 2021; Jahrens & Martinetz, 2020), and graph neural networks (Wang et al., 2020). These discriminative models focus on extracting visual features from panels to predict the correct answer.

A significant concern with the discriminate approaches, however, is whether they truly perform abstract reasoning or merely exploit dataset-specific shortcuts (Hu et al., 2021). In response, several works focused on generative models, which aim to capture the structural patterns of the RPM task rather than simply making predictions. For instance, Pekar et al. (Pekar et al., 2020) utilized a Variational Autoencoder to generate embeddings for answer panels. Shi et al. (Shi et al., 2024) proposed a deep latent variable model that encodes image attributes as latent concepts, generating answer images by applying abstract atomic rules to these concepts.

**Understanding Generalization in diffusion model**    One mystery for diffusion models is that, when the score matching loss is minimized for a finite set of training data, and the score function is learned "perfectly". Then the score function should generate samples according to a mixture of delta, i.e. memorizing training data. However, this doesn't usually happen in reality (Gu et al., 2023). Besides, multiple works have observed consistency between different diffusion models trained on the same data, suggesting diffusion models are approaching some "universal" distribution which is

not the delta mixture Zhang et al. (2023). Many have theorized about the source of generalization Kadkhodaie et al. (2023); Han et al. (2024). Some suggest the inductive bias of neural networks smooth the score function like a kernel regressor, which leads to generalization. However, these theories tackle relatively simple forms of generalization e.g. interpolating data points. Here, we train diffusion models on a dataset governed by abstract rules with unseen rules, so if it can generalize and generate according to underlying rules, it will challenge the current theory of what diffusion can learn and how it generalizes.

## 3 METHOD

### 3.1 GENRAVEN DATASET

We introduce the GenRAVEN dataset, comprising RPMs associated with 40 relational rules. Each RPM features a 3×3 matrix of panels, where three panels in each row follows a unique relational rule (Fig. 1A). To focus on rule learning and reasoning, we abstracted away the visual aspect of the sample, encoding each object with three integers representing its shape, size, and color, with $(-1, -1, -1)$ representing empty positions. Each attribute has a discrete set of allowed values, 0-6 for shape and 0-9 for size and color. Taking together, each sample RPM is represented by an integer array of shape $3 \times 9 \times 9$, which is the target of generative modeling (Fig. 1B).

Each rule is composed of an abstract relation (constant; progression ±2, ±1; arithmetic ±; XOR; OR; AND) applied to an attribute (shape, size, color, number, position). The dataset is designed so that the rule governing each row remains ambiguous when examining the first two panels and only becomes evident when all three panels are considered. This design ensures that the rule governing the row cannot be directly deduced from the first two panels alone and the model must reason the entire matrix. To study the generalization of abstract relations such as constant to new attributes, we held out 5 rules (Fig. 1D) during training. Using this set up, we can study whether model can learn to generate samples with "constant color", by observing samples with "constant size" and "progression color".

### 3.2 DIFFUSION MODEL

The diffusion models have been a prominent approach for generative modeling (Dhariwal & Nichol, 2021b) in continuous domains. It takes a holistic approach in modelling data: it learns a vector field in the data space with a function approximator $s_\theta(x, \sigma)$, which approximates the gradient of smoothed data distribution (Song & Ermon, 2019). During sampling, when samples flow along this vector field, they will be transported from a base distribution e.g. Gaussian into the target distribution $p(X)$. This vector field is learned via denoising score matching with a neural network (Hyvärinen, 2005).

**Architecture** Given the spatial structure of the task, we treated each RPM as a $9 \times 9$ image with 3 channels and adapted existing diffusion models for image generation. Specifically, we experimented with two architectures, UNet (Karras et al., 2022a) and Diffusion Transformer (DiT) (Peebles & Xie, 2023). UNet is a Convolutional Neural Network (CNN) based backbone, designed to extract information at multiple resolutions, with self-attention modules at each resolution. For a long time, it has been the default backbone for diffusion models for images (Ho et al., 2020; Dhariwal & Nichol, 2021a; Rombach et al., 2022). For our task, we adapted its architecture to match the $3 \times 3$ panel structure of the RPM samples: we changed its filter size to 3 and downsampling ratio to 3. DiT is a recent transformer-based backbone adapted for diffusion modelling (Peebles & Xie, 2023). For our task, we treated each object in RPM as a token with three features (patch size = 1), totaling 81 tokens. We also tested SiT model (Ma et al., 2024), which had the same network architecture as DiT, but based their training and inference on the framework of Stochastic Interpolant (Albergo et al., 2023).

**Sampling and in-painting** We treated the integer attributes in RPM samples as continuous values and normalized them by mean and std during training. After diffusion sampling, we denormalized and rounded the generated attribute values to the closest integer. By default, we used deterministic samplers to generate samples for efficiency: Heun's 2nd order sampler for UNet model (Karras et al., 2022a); DDIM sampler for DiT (Song et al., 2020); Runge-Kutta sampler(`dpori5`) for SiT (Ma et al., 2024). Leveraging the in-painting capability of diffusion models (Lugmayr et al., 2022; Wang

et al., 2022), we also challenged them with the generative RPM tasks: given the first 8 panels, let the model fill in the missing panel.

**Training** For all models, we used their default training parameters (detailed in Sec. A.2). We trained 1M gradient steps with mini-batch size 256, without data augmentation. All training was conducted on a single A100 or H100 GPU, which finished around 1 to 3 days.

### 3.3 EVALUATION CRITERION

One benefit of having a distribution defined by explicit rule is that, we can evaluate the generated samples precisely. For unconditional generation, there is not a sense of accuracy, so we evaluated the internal consistency of generated samples. For each row in a generated sample, we inferred the set of applicable rules. If any rule from the 40 rule set applies, the row is called *valid*. For each sample, we examine whether the same rule applies to three rows in a sample, which we call the sample rule consistent (*C3*). Similarly, if only two rows have the same rule applies, we call it *C2*. These metrics quantify the self-consistency within the generated sample.

For conditional generation, we selected 50 new samples per rule, unseen during training, as our test cases. We removed their final panel, and let the model generate the missing panel given the other 8 panels. We evaluated the of completions where the completed panel allows the same rule to apply to all three rows (*C3*), as panel completion accuracy.

## 4 DIFFUSION MODELS LEARN UNDERLYING RULES FROM SAMPLES

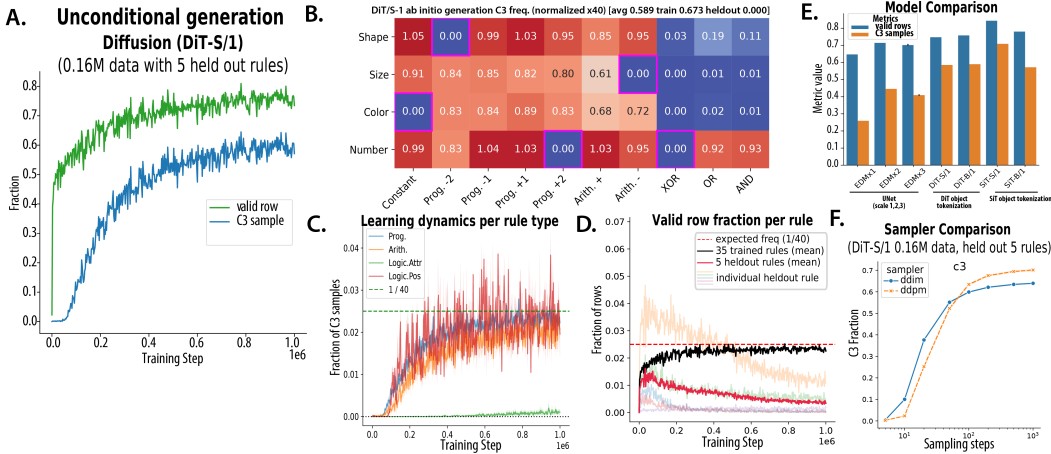

Figure 2: **Diffusion model learns to generate structurally consistent samples according to rules.** **A.** Metric of structural consistency (valid row, C3) of generated samples during model training. **B.** Frequency of generating C3 samples of each rule (showing the value ×40 to normalize) for DiT-S/1. Magenta frames showing the 5 rules held out from generative model training. **C.** The learning dynamics of generating C3 samples for each rule type. **D.** The learning dynamics of generating rows consistent with trained and held-out rules. **E.F.** Comparison of sample consistency (valid, C3) across **E.** models and **F.** samplers.

### 4.1 DIFFUSION MODEL LEARN TO GENERATE ACCORDING TO RULES WITHOUT MEMORIZATION

**Diffusion learns coherent unconditional generation better than autoregressive model** Through diffusion training, the validity of generated rows and rule consistency of the generated samples both increased (Fig. 2A). Using DiT-S/1 as our running example, after 1 million training steps, 74.8% (90% CI [74.4,75.1]%, same below) of the rows are valid (i.e. have at least one rule applying to it), and 58.5% ([57.9,59.2]%) of the samples had a consistent rule applying to all three rows (C3). This is substantially higher than the chance level (0.03% for C3; 16.9% for valid row) per independent, uniform random sampling of attribute values. Importantly, among the 200k generated RPMs, not a single RPM sample or row exists in the training dataset, which suggests that the

model did not memorize specific training examples but potentially learned the underlying rule structure. To highlight this result, we compared them to GPT2 models trained on sequentialized RPM data (details in Sec. A.3). Even doubling the width and depth of the transformer, GPT2-M performed much worse than DiT-S in generating consistent samples unconditionally (Tab. 1), with higher memorization rate (copying 15.2% rows and 3.5% samples). So what have diffusion models actually learned, how abstract is the rule they learned? We will dissect this phenomenon in Sec. 6.

**Diffusion model struggled to learn logical operation over attribute sets**  Diving into individual rules, we found diffusion models rarely generate consistent samples with Logic-Attribute type rules (Fig.2B). These rules apply logical operation (AND, OR, XOR) onto sets of attribute values. For example, the rule Shape-XOR, means when the first panel contains circles and triangles, the second panel contains triangles and squares, then the third panel could only contain circles and squares. During training, these rules were learned at a much slower rate than the other rule types (Progression, Arithmetic, and Logical operation on object position)(Fig.2C). This failure pattern is consistent across diffusion models. So why logical operation over attribute set is hard for diffusion? We would dissect the potential reason for this in Sec.6.

Table 1: **Model Comparison. valid**: fraction of valid rows. **C3,C23**: Fraction of samples where the same rule applies to three rows, or *at least* two rows; **row_mem,panel_mem**: fraction of rows or panels appearing in training set. Trained on 35 rules with 4000 samples per rule.

| Model | valid | C3 | C23 |
|---|---|---|---|
| EDMx1 | 0.647 | 0.259 | 0.493 |
| EDMx2 | 0.714 | 0.446 | 0.663 |
| EDMx3 | 0.704 | 0.410 | 0.648 |
| DiT-S/1 | 0.748 | 0.585 | 0.720 |
| DiT-B/1 | 0.758 | 0.590 | 0.733 |
| SiT-S/1 | 0.844 | 0.709 | 0.835 |
| SiT-B/1 | 0.781 | 0.572 | 0.748 |
| GPT2-S | 0.444 | 0.143 | 0.286 |
| GPT2-B | 0.519 | 0.234 | 0.389 |
| GPT2-M | 0.624 | 0.345 | 0.536 |

**Generation of held-out rules follow non-monotonic learning dynamics**  Beyond the rules seen in the training process, could diffusion models generate samples or rows consistent with the held-out rules? On the sample level, we found models rarely generate C3 samples according to held-out rules; however, on the row level, the model extended to generate rows that align with held-out rules more frequently than the chance would allow (Fig. 2D). Further, initially, the model learned to generate both trained and held-out rules with similar learning dynamics; over time, the frequency of generating held-out rules decreased while that for trained rules increased (Fig. 2D). This suggests that early on the diffusion models can "over-generalize" in a meaningful way, and later they learn to constrain the extent of generalization to focus on the trained rules.

Intriguingly, we will see the capability of panel completion according to held-out rules also follow non-monotonic dynamics, peaking at similar steps 0.1M (Fig. 3D). This highlights the model somewhat "grasped" the held-out rules initially, but then unlearned them with further training.

**Stochastic samplers improve rule consistency at higher sampling steps**  For each model family, we varied the sampler type (deterministic vs stochastic) and the number of sampling steps to see how they affect the rule consistency of unconditional generation. Across UNet and DiT, we observed that, with a small number of sampling steps, deterministic samplers can rapidly reach a comparable or higher rule consistency than stochastic samplers. However, the deterministic samplers plateaued earlier and with a larger sampling steps budget, stochastic samplers reached a higher consistency asymptotically (Fig. 2F, Fig.8). This trend is consistent with the observations on FID score for image diffusion models from Karras et al. (2022b) (Fig. 4), suggesting stochastic samplers are also more accurate and robust for discrete distribution governed by rules.

**Stochastic interpolants excel in rules learning among model families**  Comparing across model families, SiT generally generates samples with higher rule consistency than DiT, which is higher than EDM (Fig.2E, Tab.1), none of them memorize row or sample from training set. Regarding model architectures, transformer-based backbone (DiT,SiT) with object tokenization (SiT/DiT-S/1) achieved higher rule consistency than the UNet across scales (EDM-x1 to EDM-x3). It also has better rule consistency than DiT with panel tokenization (DiT-S/3) (Tab.4). This suggests the importance of object-level self-attention for rule inference. Scaling up the capacity of UNet model improved sample consistency up some level (EDMx2), while scaling up the capacity of DiT and SiT model did

not or even hurt performance. The full comparison of rule validity and consistency across network architecture and scale is shown in Fig. 2E and Tab. 1.

**Generative model learning dynamics followed rule hierarchy**   The data generating process of RAVEN has a hierarchical structure (Sec.A.1), similar to a context-free grammar system. To conform to a relational rule, the generated samples also need to satisfy the following hierarchical criteria: 1) **Object validity**: the attribute values at each location must fall within predefined ranges or be (-1,-1,-1). 2) **Row-level rule validity**: across all three panels in a row, one attribute must satisfy a rule. 3) **Cross-row consistency**: The rules applied to the three rows are the same. Fig. 7 shows the fraction of generated samples that meet each of these criteria through training. We observe a hierarchy of learning order, where the model initially learns more local data structures and subsequently adapts to more global ones. In a sense, the model first learns what are valid local elements (e.g. words), before learning to build valid combinations of them (e.g. valid phrases).

## 4.2 DIFFUSION MODEL SHOWS RULE REASONING CAPABILITY THROUGH IN-PAINTING

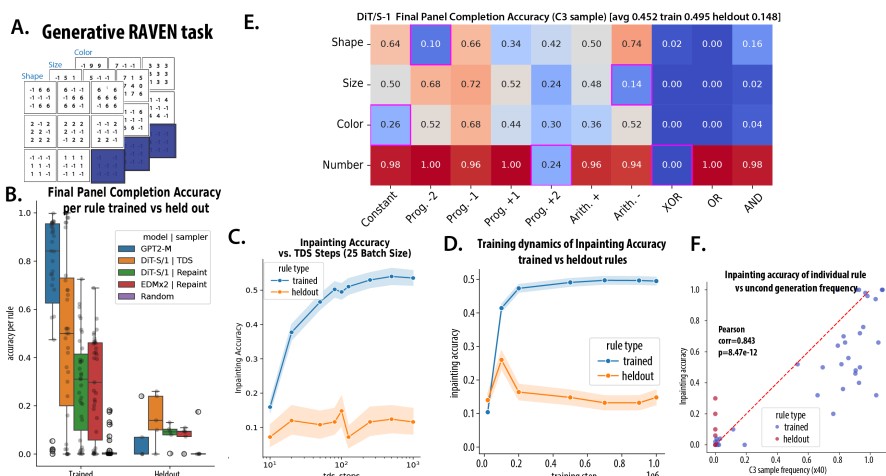

Figure 3: **Diffusion model shows reasoning capability via inpainting**. **A.** Schamtics of the GenRAVEN task. **B.** Inpainting accuracy as a function of model and sampling method. **C.D.** Inpainting accuracy as a function of **C.** TDS sampling steps and **D.** diffusion training steps, shaded errorbar = standard error (SE). **E.** Inpainting accuracy separated by individual rules. **F.** Correlation of inpainting accuracy for individual rule vs their unconditional generation frequency (C3 samples).

We've shown diffusion models learn to generate structural consistent RPM samples unconditionally, so they seem to fit the joint distribution of the rules well. Next, we want to see whether they can solve the GenRAVEN reasoning task i.e. complete missing panel according to rules using conditional sampling techniques.

**Visual reasoning as inpainting**   The generative RAVEN's task can be naturally framed as a conditional sampling problem or inpainting for image diffusion models: sample the missing panels given the values of observed panels. We found that inpainting accuracy critically depended on the sampling algorithm we used. Popular image inpainting algorithms such as Repaint and DDNM (Lugmayr et al., 2022; Wang et al., 2022) leverage the heuristics of fixing the observed dimensions of the sample, and only diffusing the masked regions. Surprisingly, these methods yield low accuracy (Fig. 3B). We found the recent method of twisted diffusion sampler (Wu et al., 2024), which leverages the twisted sampling technique from Sequential Monte Carlo (SMC) literature and is asymptotically exact, performed much better at sampling missing panels that are consistent with the rest: Of the 2000 test cases, 45.2% of panel completions were C3 consistent with existing panels (49.5% for trained rules and 14.8% for held out rules). Further, a higher sampling budget gave rise to higher accuracy (Fig.3C, Fig.9). This shows that the heuristic methods might seem to work fine in common image inpainting tasks, given the relatively loose evaluation criterion of visual inspection or classification. But for our generative RPM task which was evaluated at much higher precision, they fell short.

When comparing to autoregressive models, GPT2-M had a higher panel completion accuracy (62.4%, 70.4% for trained, 6.2% for held out) than the diffusion models even with optimized sampler (Fig.3B). Note that the inpainting performance on the held-out rules was still higher than the GPT2 and random control by uniformly sampling attributes, showing that the prior learned by diffusion model can indeed generalize beyond the trained rules to some extent.

**Panel completion capability correlates with unconditional generation capability**    During training, the capability of filling in missing part emerges at a similar timeline as the unconditional generation accuracy (Fig.3D). Notably, the inpainting accuracy for held-out rules also followed non-monotonic dynamics similar to the generation frequency in unconditional generation (Fig.2D).

Compared across rules, the inpainting accuracy was also poor for rules that were not learned well during training (e.g. logical operation on sets of attributes), (Fig.3E). Specifically, the panel completion accuracy of each rule strongly correlates with the frequency of C3 samples in unconditional generation (Pearson correlation 0.843, $p = 8.4 \times 10^{-12}$ for DiT-S/1, Fig.3F).

### 4.3 DIFFUSION MODEL LEARNED TO CLASSIFY ABSTRACT RULES VIA EMERGENT REPRESENTATION

Given that diffusion models learned to generate and complete according to rules, does it "recognize" rules and how does it represent rule internally? We trained linear probes to find representations in models that have rule classifiability.

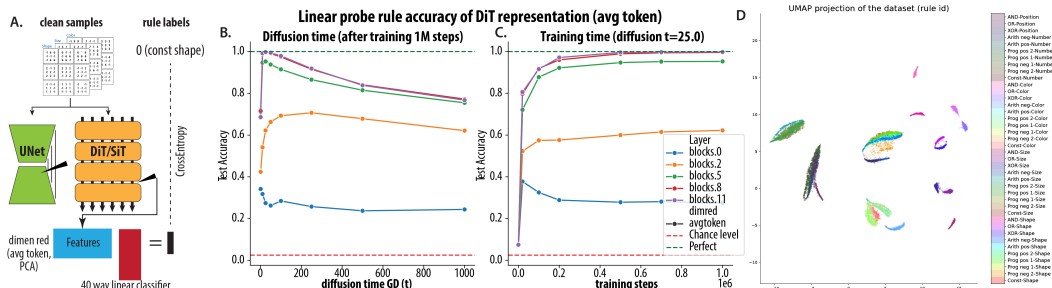

Figure 4: **Rule classifying capability emerges in diffusion model training**. **A.** Schematics of the linear probe method. **B.** Linear probe rule classification accuracy of each layer as a function of diffusion time after 1M training steps. **C.** Linear probe accuracy as a function of training steps. (diffusion time $t = 25$) **D.** UMap of token average embedding of rules from DiT-S/1.

**Linear probe method**    We passed unseen noise-free samples of the 40 rules through diffusion models with a time input and recorded activations (Fig.4). Dimension reduction (token averaging or PCA) was applied to the representation. A 40-way linear classifier was trained with 3000 training and 1000 test samples per rule, using cross-entropy loss and Adam optimizer. We systematically measured the test accuracy (rule classifiability) as a function of layer, diffusion time input, and training time for the three classes of models (details in Sec. A.5).

**Rule representation emerge**    For DiT, we found the hidden states from the last few transformer blocks are capable of rule classification. Comparing across layers, the rule classifiability increased through depth (Fig. 4B), saturating around `block.8`. Remarkably, after training, at small diffusion time ($t = 25$ out of 1000), the token-averaged hidden states at the final `block.11` reached a near-perfect test accuracy of 99.78% (with train accuracy 100%). When we reduced the size of the training set, we found that with as few as 60 samples per rule, the average token representation reached 95.5±0.2% classification accuracy (Fig.11C), highlighting the quality of the rule representation. We note that this representation is not limited to the rule types seen in the diffusion training: even for the 5 held-out rules, linear probes can read out rule identity at comparably high accuracy: trained and held-out rules both have accuracy 99.8±0.2% (Fig.11B). In the following, we will report the results with token averaging, with the PCA results in Fig.10.

Notably, the rule classifiability was significantly modulated by diffusion time or noise scale input to diffusion model. For the top layer, the rule classifying accuracy peaked at small but non-zero noise

scales ($t$ around 10-50 out of 1000 steps), and was significantly lower for both very small and large $t$. Similar results were found for the other two class of models (SiT and EDM), where the best rule classifying representation is obtained at the time input $t$ representing low but non-zero noise scale.

Geometrically, the samples of each rule form clusters in representations space, enabling linear classification, (see UMap and the representation similarity matrix in Fig.11D,E,F).

**Diffusion model learns to recognize rules before generating.** Tracking the rule classifiability as a function training time, we found the model learns to classify rules early on in training, much earlier than learning to generate consistent rules, reaching 91.6% after 0.1M optimization steps at top layer (c.f. the C3 fraction is 10.4% at 0.1M step).

We also found that the difficult rules in generation (Logic-Attribute) were also hard for recognition: for earlier layers, there was a significant gap in classification accuracy for these rules versus other rules, which is closed in the final layers (Fig.11A). This shows it took more layers to obtain separable representations for difficult rules for generation.

Similar results were found for SiT and EDM models. SiT models had peak test accuracy 99.6% (Sec.B.6.2, Supp. Fig. 12). For UNet model (EDM), rule-classifying representation were found at the bottleneck layers, with peak accuracy 93.9% (EDMx2) (Sec.B.6.3). This is in line with the common intuition that the compressed representation in bottleneck layer contains high-level information.

In summary, we found that the diffusion model obtained near perfect rule classifying capability by generative modeling RPM data. In this perspective, diffusion modeling is a powerful self-supervised learning method, and our finding extends beyond the previous results about object classifying representation emerged during diffusion training (Chen et al., 2024; Mukhopadhyay et al., 2023) to more abstract rule data. On a high level, this is also consistent with the notion that by modeling the whole data distribution, generative models can discover key factors in the true data-generating process (e.g. abstract rules), and construct a *world model*.

## 5 SCALING PROPERTIES OF DIFFUSION RULE LEARNING

How do data and model size affect rule learning in diffusion? We manipulated the scale of the training dataset from 400 samples to 400000 samples per rule, and we trained models of the three families at different scales, with the same training step (1M) and batch size (256). Hyperparameters for each model scale were detailed in Sec. A.2.

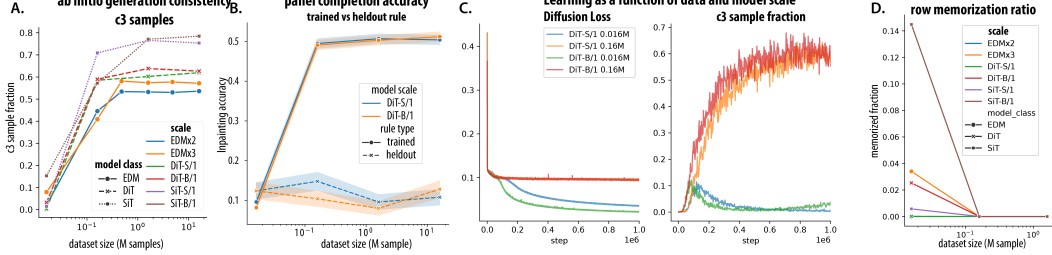

Figure 5: **Scaling property of rule learning. A.** Data scaling curve of *ab initio* sample generation consistency (C3 fraction) for Diffusion models (EDM, DiT, SiT). **B.** Scaling of panel completion accuracy with model and dataset size of DiT models. **C.** Rule learning "collapsed" at small data scale. Loss and C3 fraction during training, modulated by dataset size and model size. **D.** Row memorization ratio as a function of model and data scale.

**Rule learning emerges at a certain data scale** We found that the rule learning capability emerged at a data scale, consistent across models. When only 400 samples per rule were seen during training, all models failed to learn the rule: for unconditional generation, C3 sample fraction dropped below 15% (Fig. 5A); for conditional sample completion, the completion accuracy dropped to around 10% (Fig. 5B). Intriguingly, at low data regime, the rule consistency followed non-monotonic learning dynamics: it initially grew as in the higher data regime; then the learning "collapsed", i.e. the rule consistency of samples dropped towards zero, accompanied by the training loss rapidly dropped, "breaking through the floor" (Fig. 5C). Similar phenomena were observed consistently across the three model families (see Supp. Fig.14).

**Interaction of model capacity and the training data scale**    Are bigger models better rule learners? Across the three model families, we found larger models do not yield higher rule consistency in their generation, esp. when the amount of training examples is limited (400-4000 per rule). Further, when the data scale is small (400 sample per rule), bigger models tend to "collapse" faster (Fig. 5**A**). However, the larger models also benefited more from increased data scale (40k-400k per rule) (Fig. 5**B**). This trend is reminiscent of the classic statistical learning theory, where more data are needed to constrain a bigger model, or it will overfit and perform worse.

**Diffusion models tend to memorize more global structure at small data regime, leading to failure**    Next, we examined the memorization pattern at different data scales. At the default scale (4000 per rule), the models do not memorize rows or samples. However, at low data regime, likely due to the more repetitions over the fixed training set, a larger fraction of rows and samples were memorized esp. for models with higher capacity (SiT-B, DiT-B, EDMx3) (Fig. 15).

This could be related to the memorization phenomena in the literature: given a smaller number of training samples from a dataset, diffusion models tend to memorize the exact training sample (Yoon et al., 2023; Zhang et al., 2023). However, in our case, even when the rule learning collapsed, the model didn't simply regurgitate the training samples: the row and sample memorization fraction remained mostly lower than 10% (Fig.5D). This suggests a higher level of memorization-generalization dichotomy or model collapse.

## 6 DIFFUSION GENERALIZES BY RECOMBINING MEMORIZED PARTS

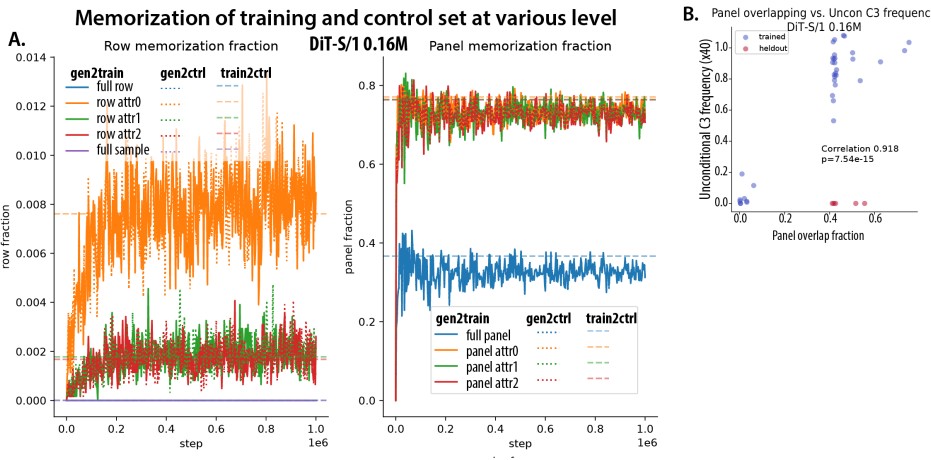

Figure 6: **Dissection of memorization and generalization A.** Memorization of training (Solid) and control (dashed :) set at multiple levels for samples generated through training. Horizontal dashed lines show the fraction of overlap between the training and control sets. **B.** Panel overlapping fraction correlates with generation frequency of each rule.

It's well known that, if the diffusion model learned the score field exactly, it should converge to the exact training data points (Gu et al., 2023). However, from our empirical results, these diffusion models can generate novel rule-conforming rows and samples. So what have diffusion models actually learned?

To answer this, we evaluated the memorization of generated samples and their parts: rows, panels, and rows and panels when considering a single attribute. We evaluated what fraction of samples and parts have exact copies in the training set for the model. As a control, we also generated another control dataset with the same number of samples per rule and computed the memorization fraction for the control set. We hypothesized that even though memorization has not happen on the row level, it might happen on local patch and attribute level. Further, the diffusion model learn to recombine these memorized local components to create new samples in a rule-conforming manner.

**Diffusion model memorize and recombine panels**    Although diffusion models have no memorization on the sample and row level, we found they had significant memorization on the panel level. At the default data scale (4000), after training, we found across all models consistently, [32-35]% of generated panels had exact copies from the training set. Interestingly, they also had a similar fraction of panels that had copies from the unseen control set (Fig.6A). This reflects the statistical property of the data-generating process: when we generate another 4000 samples per rule (held out 5 rules), we found it also has 35.2% panels that overlap with the training set (Fig.6A, horizontal line). Similar results hold for the other levels (e.g. panels per attribute). In this sense, the diffusion models are "memorizing" parts in the training set but *no more than* separately generated data "memorizing" the training set. So, we can call this *benign memorization*, or approaching the true data distribution.

**Panel reuse predicts easiness of individual rule**    Further, when we examine individual rules, some rules have more overlapping panels across random splits of datasets and some have fewer overlapping panels. We found that Logic-Attribute type rules have an especially low fraction of panel that overlap with independently generated training set (1.9% vs others 46.7%). We found that for trained rules, the panel overlapping rate of each rule highly correlates with the frequency of model generating C3 samples according to that rule, (correlation 0.918, $p < 1 \times 10^{-10}$ Fig.6B). This supports our notion that diffusion models need to memorize local parts i.e. panels to learn a certain rule successfully. In another sense, the recurring frequency of parts determines the overall difficulty of a rule.

**Panel overlap as an explanation to rule learning collapse**    We found at a small data scale (400 per rule), there will be substantially fewer panel overlaps between the training set and independently generated control set: 12.4% of panels are overlapping. We think the lack of memorizable parts might explain why rule learning collapses at this scale.

# 7    DISCUSSION

We thoroughly investigated the learning dynamics and memorization behavior of rule learning in diffusion models in the RPM task inspired by cognitive tests. We found diffusion models can learn to unconditionally generate novel samples and complete patterns according to rules. Representations emerged in diffusion models are also capable of classifying rules at near-perfect accuracy. Our results may inspire further theories about the source of generalization in diffusion models.

**Weakness of diffusion models in rule consistent panel completion**    We are surprised that the panel completion accuracy of diffusion models is generally lower than the C3 frequency of their unconditional generation, and also lower than autoregressive counterparts. As we may naively think, modeling conditional density should be easier than modeling joint distribution. One potential reason is that, even though unconditional sampling algorithms for diffusion have been studied extensively Karras et al. (2022b); Zhao et al. (2024); Liu et al. (2022); Lu et al. (2022), we still do not have the perfect method for sampling the conditional distribution from diffusion models. Just as TDS perform better than simple heuristic method e.g. Repaint, better conditional sampler may boost the reasoning capability of diffusion models further. Our results also provide evidence that for tasks that are evaluated precisely (e.g. abstract rules), a better sampler is critical.

**Improving sample complexity**    We identified one key condition (data scale) that gives rise to rule learning behavior in diffusion models. Currently, the data scale required for rule learning (4000 samples per rule) is still significantly higher than what human needs to learn the rules. Future works are required to investigate how to reduce the sample complexity of rule learning.

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

# A  DETAILED METHODS

## A.1  DATASET CONSTRUCTION

We have a procedural algorithm to generate rows following the 40 rules. The basic workflow is the following:

1. Choosing the rule following dimension (Shape, Size, Color, Number, Position);

2. Choosing the rule type (Constant, Progression, Arithmetic, XOR, OR, AND);

3. Switching based on the type of rules
    - If it's Constant, Progression, Arithmetic of object attributes (shape, size, color), then each panel will only have single value for the attribute i.e. object all have same size in one panel. We will choose three values $(X_1, X_2, X_3)$ that follow the rule (e.g. Progression +2, will be $X_1 = 1, X_2 = 3, X_3 = 5$).
    - If it's XOR, OR, AND of object attributes, then each panel will have a set of attributes, which suffice the logic operation. Decide the number of objects of each attribute value.

4. Choose the other attribute dimensions of the object.

5. Check if the other attribute follow other rules, if so, regenerate the other attributes

We synthesized 1,200,000 rows per rule. Three rows following same rule will be randomly combined into a Raven's Progression Matrix sample. These RPMs are encoded as a $3 \times 9 \times 9$ integer matrix. We used $[1.5, 2.5, 2.5]$ and $[2.5, 3.5, 3.5]$ as mean and std of the three channels, to normalize the RPM encoding tensors.

## A.2  DIFFUSION MODEL AND TRAINING DETAILS

### A.2.1  EDM/UNET

Our implementation was based on the original EDM code base (Karras et al., 2022b) and its simplified version `https://github.com/yuanzhi-zhu/mini_edm`.

For model scaling, we tested EDM models with three scales, EDMx1, EDMx2, EDMx3, which double and triple the width and depth of the UNet model. All UNet models have 3 resolution blocks $(9, 3, 1)$, at each resolution level the channel numbers are increased by a multiplier $(1, 2, 4)$. At each resolution, there are self-attention modules. Specifically, `EDMx1,EDMx2,EDMx3`, used base channel number 64, 128 and 192; while they used 1, 2, and 3 Residual Layer per resolution.

### A.2.2  DIT

Our implementation was based on the original DiT code base (Peebles & Xie, 2023), `https://github.com/facebookresearch/DiT`

For the major part of the paper, we used patch size 1, which treats each object as a token, with 81 tokens. In our ablation experiments, we also tested the model with patch size 3, which treats each panel as a token, with 9 tokens. The panel tokenization trains much faster but doesn't perform as well (see Tab.4). For model scaling, we adapted the configuration from DiT and SiT, and used their Small (-S) (12 layers, 384 hidden dimensions, 6 heads) and Big (-B) scale (12 layers, 768 hidden dimensions, 12 heads).

### A.2.3  SIT

Our implementation was based on the original SiT code base (Ma et al., 2024), `https://github.com/willisma/SiT`.

The network architecture of SiT is almost identical to DiT, with the same object tokenization and configuration for Small and Big scale. For the training configuration, we used Velocity prediction, Linear Path, and `none` loss weighting.

### A.2.4 DEFAULT SAMPLERS

By default, during model training, we used deterministic samplers to generate samples for efficiency: Heun's 2nd order sampler with 18 steps for UNet model (Karras et al., 2022a); DDIM sampler with 100 steps for DiT (Song et al., 2020); off-the-shelf Runge-Kutta sampler of order 5 (`dpori5`) for SiT (Ma et al., 2024); and we systematically evaluated the effect of samplers in Fig. 8.

### A.3 AUTOREGRESSIVE MODELS BASELINE

Here we also used auto-regressive models **GPT2** Radford et al. (2019) to solve RAVEN's task, learning the sequence of objects in an RPM. Similar to DiT, we treated each object as a token. The three integer attributes of each token were embedded through separated embedding layers with 1/3 hidden dimensions and then concatenated as the token embedding. To reflect such latent space structure, the hidden state outputs from the transformer or Mamba were chunked into 3 parts and decoded separately into attributes of the next token. We used the default positional encoding for GPT2 and no position encoding for Mamba. We train these models with the next token prediction objective. A key design decision is an order to sequentialize RPM data into 1d. Here we first scan the objects within each panel, and then follow the raster order of panels (panel1 in row 1, P2 row1, ...P3 row 3), forming a sequence of 81 tokens. We prepend the same starting token `[SOS]` for each sample. For unconditional generation, we started from `[SOS]` and autoregressively sampled 81 tokens with temperature 1.0. For panel completion, we started from `[SOS]` and 72 existing tokens (corresponding to first 8 panels), and sampled 9 tokens with temperature 1.0.

### A.4 IMAGE CONDITIONAL SAMPLING METHOD (INPAINTING)

We used the Twisted Diffusion Sampler based on their official implementation `https://github.com/blt2114/twisted_diffusion_sampler`. We compared that to a custom implementation of the Repaint method combined with Heun (for EDM) and DDIM sampler (for DiT).

### A.5 LINEAR PROBE OF LATENT REPRESENTATION

We forwarded noise-free samples of the 40 rules (unseen in diffusion training) through diffusion models with a specific time input, then record activations from various layers (Fig.4). For each rule, we used 3000 samples as the training set and 1000 samples as the test set. We trained a 40-way linear classifier to classify rules from the representation with cross-entropy loss (as opposed to One-vs-All binary classification). We fit the classifer with 5000 full-batch gradient descent steps using Adam optimizer with learning rate 0.005. We report the test set accuracy after fitting. Since the activation could be high dimensional, complicating classifier training, we employed various dimension reduction techniques: 1) token averaging: average the activation across space (UNet) or tokens (DiT, SiT); 2) PCA: concatenate the activation across space or tokens and then project activation onto the top $K$ PCs. The dimension-reduced features are standardized with zero mean and unit standard deviation before sending into linear classifier.

## B    EXTENDED RESULTS

### B.1    HIERARCHICAL LEARNING DYNAMICS IN RULE LEARNING

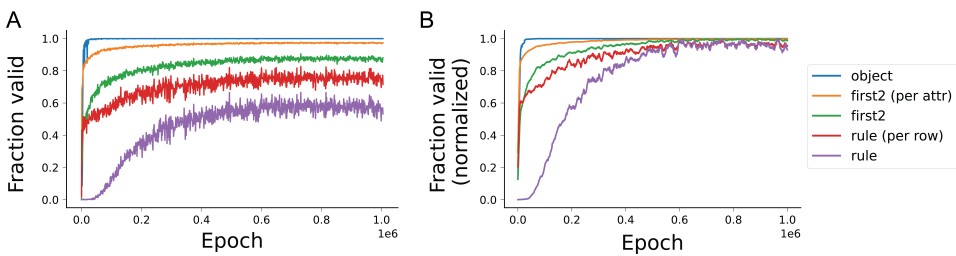

Figure 7: **Hierarchy of rule learning dynamics for DiT-S/1 model.** Generated samples across training evaluated by different criteria from local to global validity, i.e. validity of object, first two panels per attribute, first two panels considering all 3 attributes, valid rule per row, consistent rule across three rows. **A.**. Raw fraction **B.** Fraction normalized by the max in the trajectory.

### B.2    SAMPLER COMPARISON FOR UNCONDITIONAL GENERATION

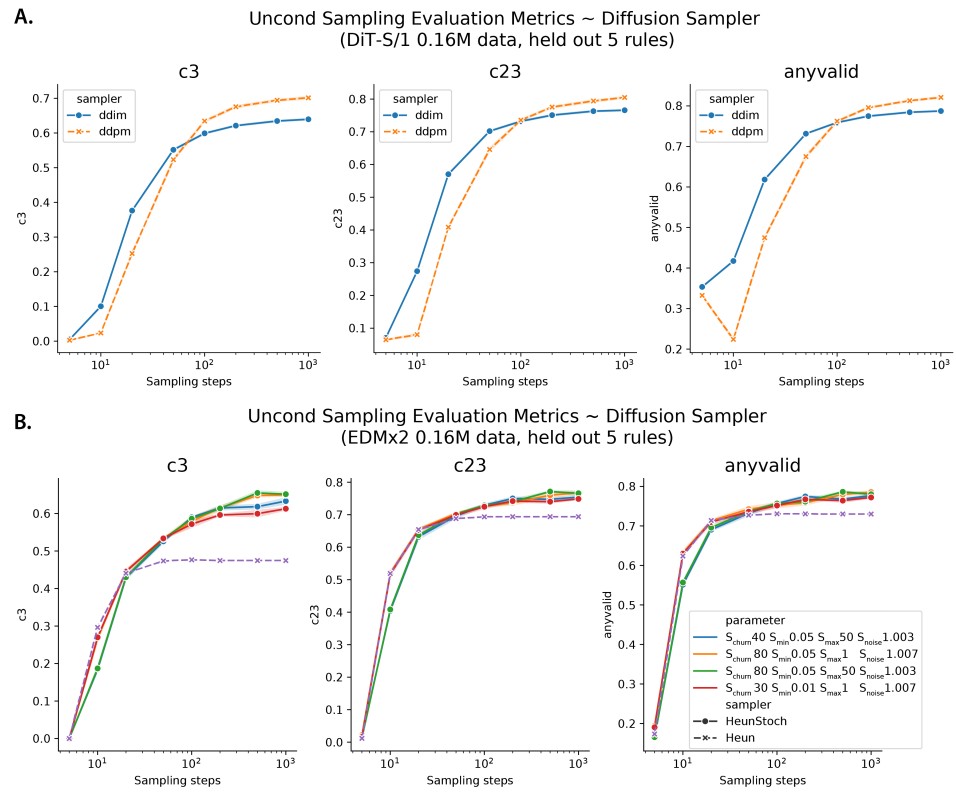

Figure 8: **Comparison of stochastic and deterministic diffusion sampler in unconditional generation**. **Left**, C3 sample fraction; **Middle** C2+C3 sample fraction, **Right**, valid row fraction. **A.** DiT-S model. **B.** EDMx2 model

### B.3 SAMPLER COMPARISON FOR CONDITIONAL GENERATION (INPAINTING) IN DIFFUSION MODELS

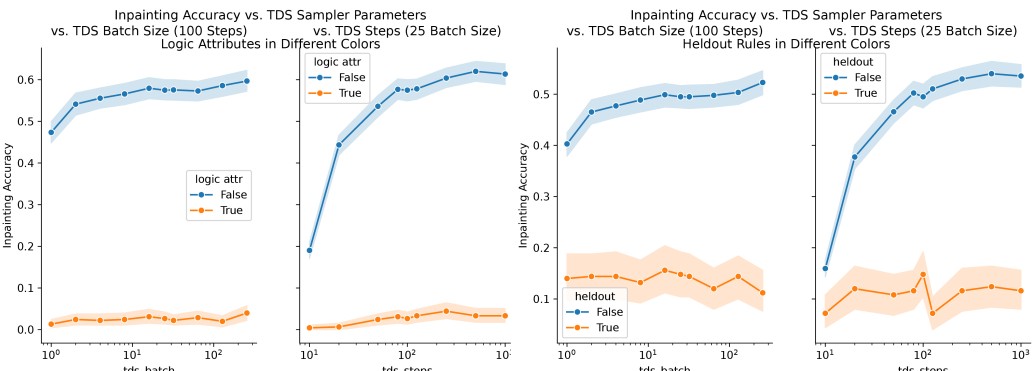

Figure 9: **Parameter tuning for Twisted Diffusion Sampler**. Left column, population size in TDS. Right column, sampling step number in TDS. Separated by hard (Logic-Attribute) vs easy (Others); or trained vs heldout.

## B.4 PERFORMANCE AND MEMORIZATION PROFILE OF DIFFUSION MODELS

Table 2: **Performance and memorization comparison across models (5 held outs).**
**Performance metrics.valid**: fraction of valid rows. **C3,C23**: Fraction of samples where the same
rule applies to three rows, or *at least* two rows;
**Fraction of memorization at various levels**. *row A0* means row memorization ratio considering
only attribute 0 in it. Same for *pan A0*.
All models were trained with 5 held out rules, with 4000 samples per rule.

| Model | valid | C3 | C23 | sample | row | row A0 | row A1 | row A2 | panel | pan A0 | pan A1 | pan A2 |
|---|---|---|---|---|---|---|---|---|---|---|---|---|
| EDMx1 | 0.647 | 0.259 | 0.493 | 0.000 | 0.000 | 0.004 | 0.001 | 0.001 | 0.328 | 0.736 | 0.742 | 0.729 |
| EDMx2 | 0.714 | 0.446 | 0.663 | 0.000 | 0.000 | 0.007 | 0.002 | 0.001 | 0.346 | 0.751 | 0.752 | 0.744 |
| EDMx3 | 0.704 | 0.410 | 0.648 | 0.000 | 0.000 | 0.007 | 0.002 | 0.002 | 0.343 | 0.743 | 0.747 | 0.747 |
| DiT-S/1 | 0.748 | 0.585 | 0.720 | 0.000 | 0.000 | 0.008 | 0.002 | 0.002 | 0.325 | 0.746 | 0.724 | 0.719 |
| DiT-B/1 | 0.758 | 0.590 | 0.733 | 0.000 | 0.000 | 0.008 | 0.002 | 0.002 | 0.322 | 0.727 | 0.724 | 0.732 |
| SiT-S/1 | 0.844 | 0.709 | 0.835 | 0.000 | 0.000 | 0.010 | 0.001 | 0.001 | 0.353 | 0.759 | 0.751 | 0.748 |
| SiT-B/1 | 0.781 | 0.572 | 0.748 | 0.000 | 0.000 | 0.010 | 0.002 | 0.002 | 0.333 | 0.735 | 0.745 | 0.739 |

Table 3: **Performance and memorization comparison across models (All).**
**Performance metrics.valid**: fraction of valid rows. **C3,C23**: Fraction of samples where the same
rule applies to three rows, or *at least* two rows;
**Fraction of memorization at various levels**. *row A0* means row memorization ratio considering
only attribute 0 in it. Same for *pan A0*.
All models were trained on all 40 rules, with 4000 samples per rule.

| Model | valid | C3 | C23 | sample | row | row A0 | row A1 | row A2 | panel | pan A0 | pan A1 | pan A2 |
|---|---|---|---|---|---|---|---|---|---|---|---|---|
| EDMx1 | 0.642 | 0.233 | 0.467 | 0.000 | 0.000 | 0.004 | 0.001 | 0.001 | 0.359 | 0.753 | 0.760 | 0.737 |
| EDMx2 | 0.736 | 0.480 | 0.690 | 0.000 | 0.000 | 0.007 | 0.002 | 0.002 | 0.372 | 0.759 | 0.758 | 0.758 |
| EDMx3 | 0.723 | 0.458 | 0.681 | 0.000 | 0.000 | 0.008 | 0.002 | 0.001 | 0.373 | 0.759 | 0.758 | 0.753 |
| DiT-S/1 | 0.754 | 0.603 | 0.726 | 0.000 | 0.000 | 0.008 | 0.002 | 0.002 | 0.351 | 0.741 | 0.742 | 0.735 |
| DiT-B/1 | 0.765 | 0.611 | 0.743 | 0.000 | 0.000 | 0.009 | 0.002 | 0.002 | 0.343 | 0.741 | 0.734 | 0.730 |
| SiT-S/1 | 0.846 | 0.722 | 0.843 | 0.000 | 0.000 | 0.011 | 0.002 | 0.001 | 0.380 | 0.766 | 0.761 | 0.758 |
| SiT-B/1 | 0.792 | 0.611 | 0.768 | 0.000 | 0.000 | 0.011 | 0.001 | 0.002 | 0.373 | 0.763 | 0.756 | 0.755 |

## B.5 Ablation on model architecture

Table 4: **Ablation of tokenization scheme for DiT. valid**: fraction of valid rows (conforming to any rule). **C3, C2**: Fraction of samples where the same rule applies to three rows or two rows; **row_mem, panel_mem**: fraction of rows or panels appearing in training set. **DiT-S/1, DiT-B/1**: DiT with 1x1 object tokenization **DiT-S/3, DiT-B/3**: DiT with 3x3 panel tokenization.

| model | valid | C3 | C2 | row mem | panel mem |
|---|---|---|---|---|---|
| DiT-S/1 | 0.739 | 0.551 | 0.159 | 0.000 | 0.320 |
| DiT-B/1 | 0.683 | 0.436 | 0.198 | 0.000 | 0.325 |
| DiT-S/3 | 0.659 | 0.334 | 0.220 | 0.000 | 0.331 |
| DiT-B/3 | 0.643 | 0.241 | 0.258 | 0.000 | 0.345 |

**Object tokenization helps rule learning compared to panel tokenization for DiT model**

## B.6 Emergent rule representation in Diffusion models

### B.6.1 Representation in DiT models

The time input of DiT model follows the same convention as the ancient Gaussian Diffusion code base, where time is discretized into 1000 steps, with $t = 0$ being the clean image and $t = 1000$ being the pure noise. Here we found the best rule classifying representation around $t = 25$, at the top transformer block, at the end of training.

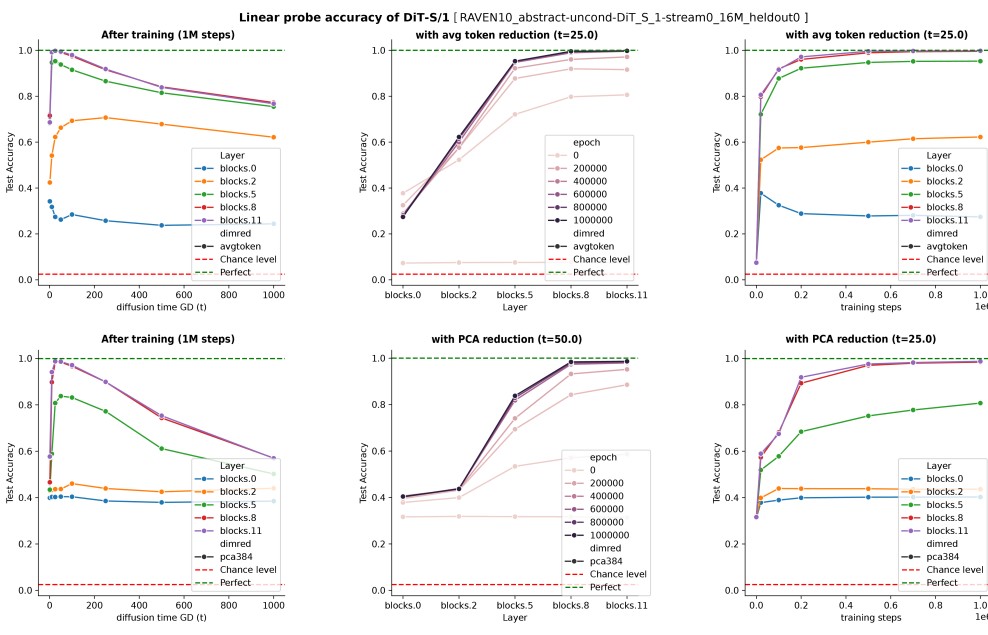

Figure 10: **Comparing Rule classifying capability with token averaging vs PCA reduction**. Linear probe accuracy is plotted as a function of diffusion time (left), layer depth (middle), and training time (right). **Upper row**: Linear probe fit on average token representation. **Lower row**: Linear probe fit on PCA of concatenated latent states (with 384 pc dimensions).

### B.6.2 Representation in SiT models

SiT model enjoys the same network architecture, but was trained with different conventions and frameworks (Stochastic interpolants). The most noticeable difference is the convention of time input $t$ in SiT, where $t = 0$ is pure noise and $t = 1$ is the clean image, and the sampling process integrates $t$ in the positive direction. Thus, most results are comparable to DiT, as long as we remap the time input

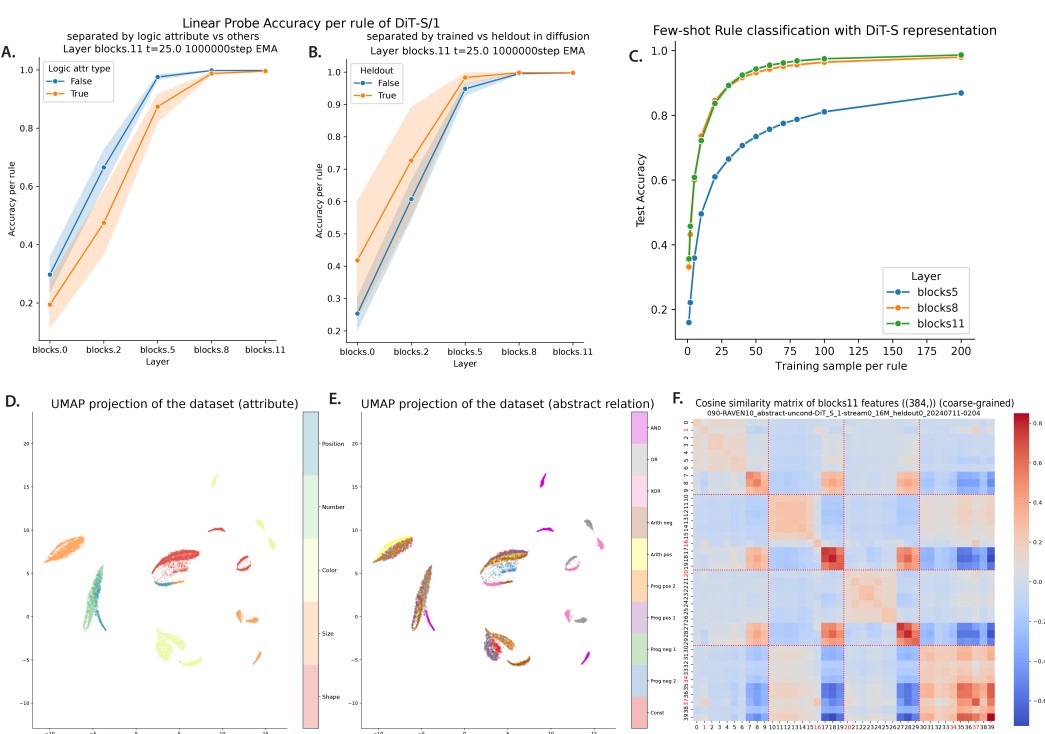

Figure 11: **Detailed analysis of emergent rule representation in DiT-S/1 (average token)**.

**A.** Test accuracy of rules across the depth of DiT, separated into logic-attribute rules versus other rules. Test accuracy of logic-attribute rules saturates at deeper layers, showing they are harder also for generation.

**B.** Test accuracy of rules across the depth of DiT, separated into the 35 trained rules versus the 5 heldout rules in diffusion training.

**C.** Test accuracy as a function of training data points and depth, showing that as few as 50 training samples per rule are enough for learning the linear classifier.

**D.E.** UMAP of representation of test RPM samples, colored by attribute dimension (**D.**) and abstract relations (**E.**) of corresponding rule.

**F.** Coarse-grained representation similarity analysis of RPM samples, 100 samples per rule, each element in the matrix represents the average cosine similarity between $100 \times 100$ pair of samples per rule pair.

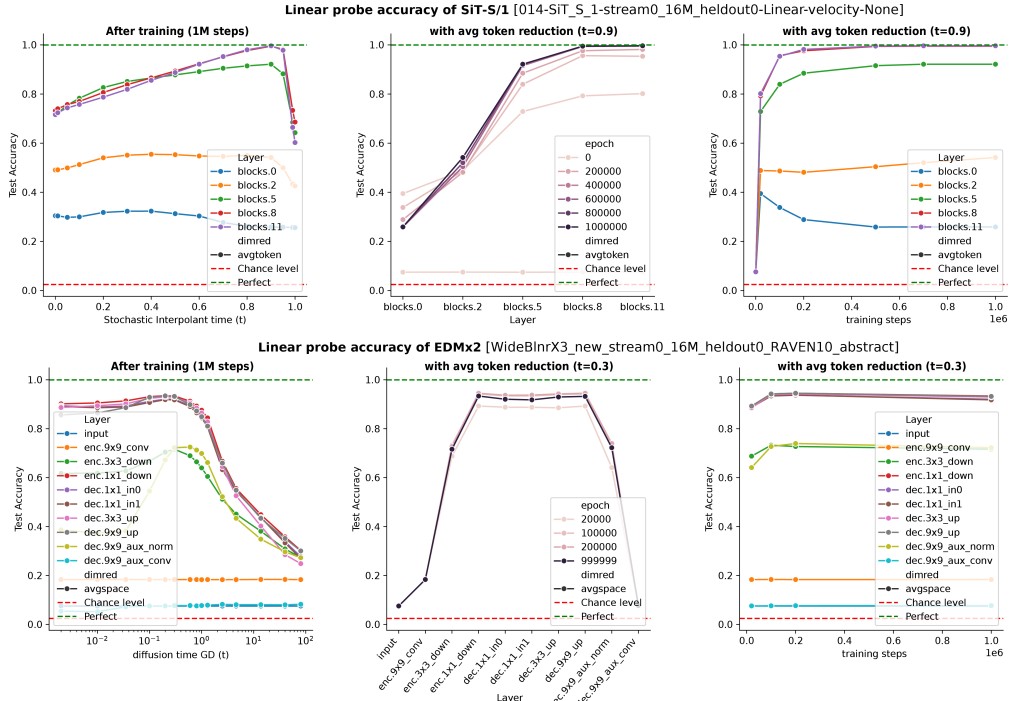

Figure 12: **Rule classifying capability emerges in diffusion model training, SiT, EDM**. Linear probe accuracy is plotted as a function of diffusion time (left), layer depth (middle), and training time (right). **Upper row**: SiT. **Lower row**: EDM.

$t$. Here we found, at $t = 0.9$, the linear probe test accuracy was also almost perfect 99.6% for the token averaged hidden states at the final transformer block, comparable to DiT model (Fig. 12 **Upper Row**). At this diffusion time $t = 0.9$, the rule accuracy also increased through depth, saturating around the last few transformer blocks, similar to DiT. Similarly, this rule classifying representation was learned rapidly during training.

### B.6.3 REPRESENTATION IN EDM MODELS

For EDM models, the linear probe accuracy also depends on the diffusion time or noise scale $\sigma$. Here the accuracy peaked around $\sigma = 0.3$ (with $\sigma_{min} = 0.002, \sigma_{max} = 80.0$). The bell-shaped dependency also follows the same trend as in DiT and SiT, where the best rule classifying representation is obtained at the time input $t \approx 0.3$ representing a low but non-zero noise scale. Since EDM (UNet) model has a different architecture, the rule classifying representations emerged at the bottleneck layers. The middle layers around the bottlenecks all have similar linear classification accuracy (Fig. 12 **Lower Row**). Across training steps, EDM models' linear probe accuracy was the highest at 0.1M to 0.2M steps (peak at 94.5%), and then decayed slightly with extended training. Further, this representation was also learned early in training: Even with just 20000 training steps, EDM models acquired rule classification accuracy of 89.3%.

## B.7 SCALING OF RULE LEARNING CAPABILITY AND MEMORIZATION

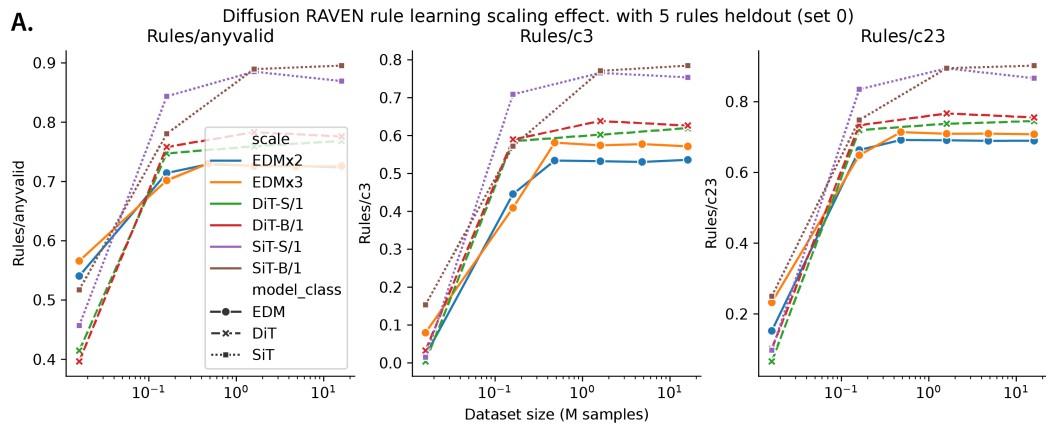

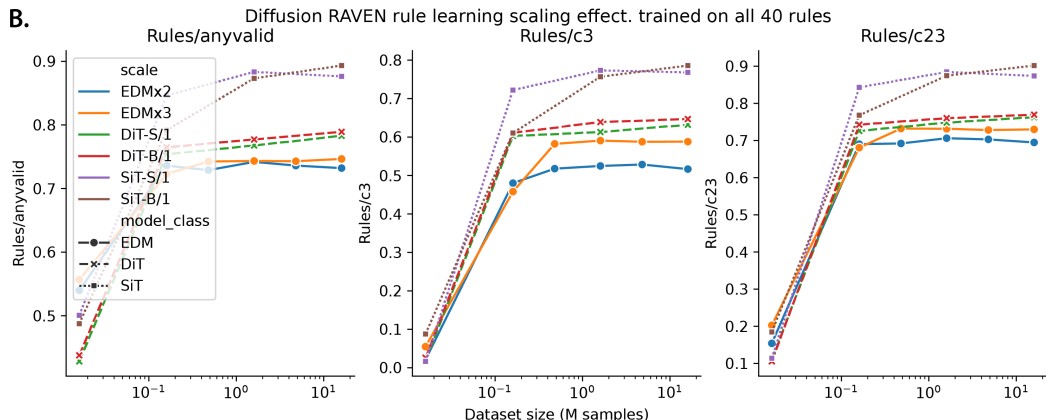

Figure 13: **Extended data scaling curves for diffusion model** Valid row fraction (left), C3 sample fraction (middle), C2+C3 sample fraction (right) as a function of dataset size and model size for EDM, DiT, SiT models. **A.** for models trained on 35 rules (with 5 heldout rules); **B.** for models trained on all 40 rules.

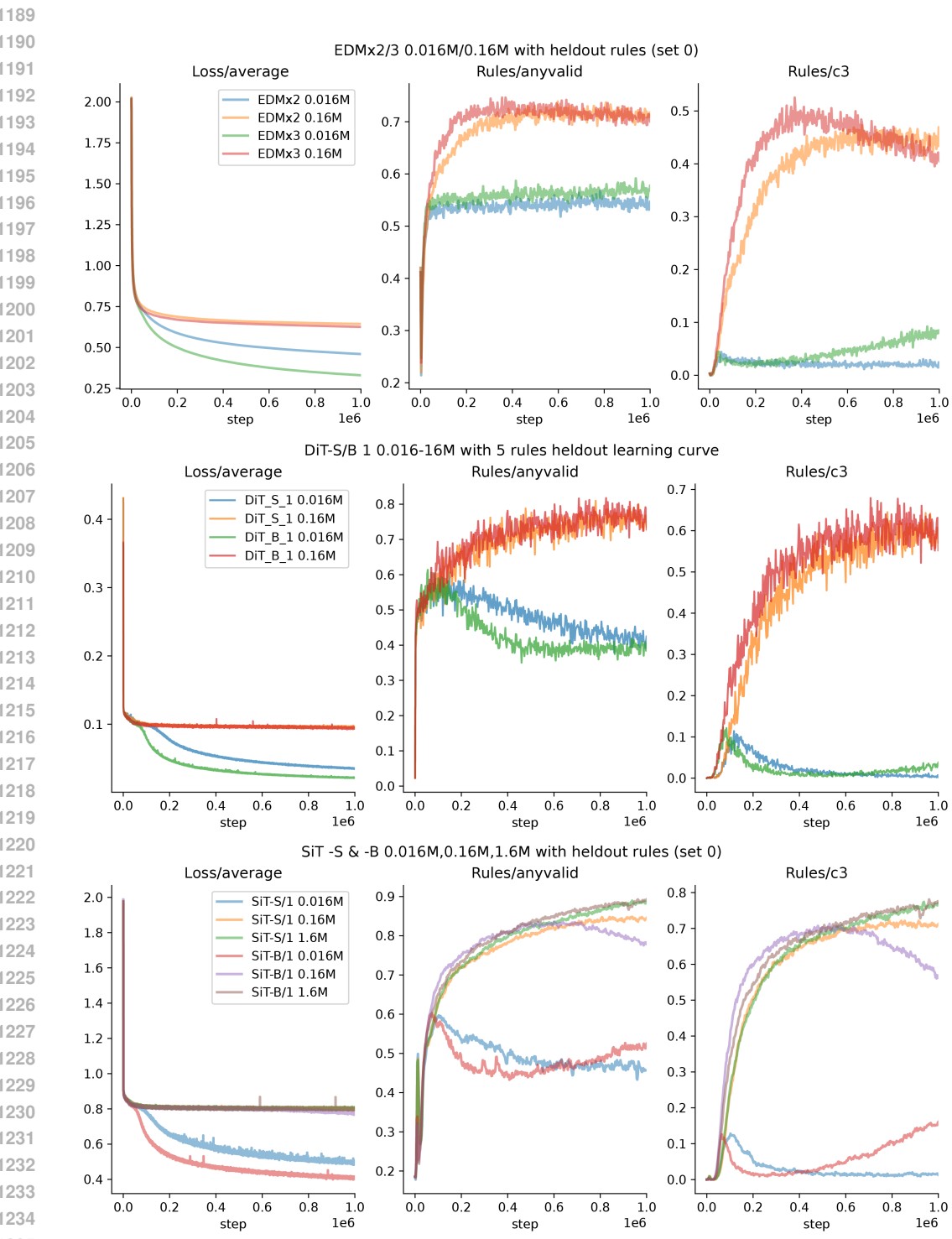

Figure 14: **Collapse of rule learning for three families of diffusion models** Training loss (left), Valid row fraction (middle), C3 sample fraction (right) as a function of training steps, modulated by dataset size and model size. Similar plots shown for the 3 families of diffusion models EDM (**A.**), DiT (**B.**), SiT (**C.**).

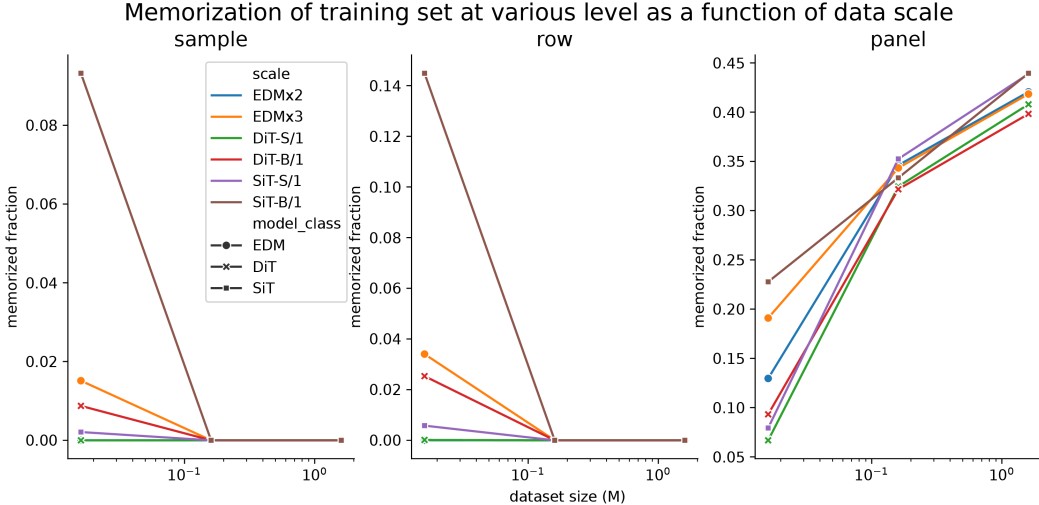

Figure 15: **Diffusion models memorized rows and samples more at low data regime** Fraction of Sample (left), Row (middle), Panel (right) as a function of model and data scale EDM (**A.**), DiT (**B.**), SiT (**C.**).

