# OpenReview forum: "How do diffusion models learn and generalize on abstract rules for reasoning?"
_ICLR.cc/2025/Conference — Submitted to ICLR 2025_

### Official Review · Reviewer_8V8a · 2024-10-27

**Soundness:** 2
**Presentation:** 3
**Contribution:** 2
**Rating:** 5
**Confidence:** 4

**Summary:**

This is experimental paper on evaluating diffusion models on a specific puzzle, named Raven’s progression matrix task. This task contains 3*9*9 features, and has 40 relational rules. The task is figure out the underlying relational rule based on the observation of two rows, and complete the third row based on the observation. This paper generates many data points for this task, and tried different variants of diffusion models, then gets some interesting observations, including the diffusion models can synthesize novel samples, learn from local parts of training samples, and can do pattern completion task for unseen rules. Moreover, the feature layer of the models contains semantic meanings of the rules.

**Strengths:**

I think the strength of the paper mainly comes from the small and concrete experimental setting. Indeed, in the Raven's task, there are limited number of parameters and rules, and one can easily control the data distribution and observe the performance of the diffusion models.

Originality: I think the main originality arise from the concrete experimental setting, which also contains logic inference as well. I think this can be treated as one of the simplest settings for reasoning.

Quality: the paper did careful investigation on various aspects of the task, with detailed discussions.

Clarity: the paper is easy to follow.

Significance: the paper is significant in the sense that it is the first paper that tries to investigate the reasoning ability of diffusion models using a small and concrete task.

**Weaknesses:**

I think the main weakness of the paper is lack of an interesting and useful conclusion that we previously do not know about diffusion models. The main body of the paper is about experimental results, but I personlly did not find the results particularly surprising. I will follow the abstract to discuss one by one:

1. "diffusion models can synthesize novel samples": this seems not so surprising, because we already know that diffusion models can synthesize novel images (like flying monkeys) not existed before.
2. "memorized and recombined local parts of the training samples": this is an interesting point, because previously we did not know how diffusion models can learn the structure. However, in this paper, they have this claim mainly because of Figure 7, which is a plot showing the model first learns local part, then global parts. It would be better if the authors can provide deeper discussions on this point.
3. "advanced sampling techniques were needed for inpainting": this is also an interesting point, but it seems that here the authors simply mean that Sequential Monte Carlo will give better performance, but did not provide any deep analysis.
4. "pattern completion capability can generalize to rules unseen": this is an interesting point, but it seems that the authors mainly provide some unseen examples in test set, and show that diffusion models can work well on these examples. The fact that models can generalize to unseen cases has been observed by many previous studies, and it would be great if the authors can use Raven's task to provide better intuitions and explanations.
5. "rule representation": this is not in particular surprising, because people previously already observe that deep networks can learn semantic features, or self-supervised learning framework can learn very strong features.

**Questions:**

I do not have additional questions, as I listed my questions in the weakness section.

---

> ### Author Response · Authors · 2024-11-25
> **Responses to the weaknesses of the paper and new experimental results regarding the generalization of rule representation**
>
> > "diffusion models can synthesize novel samples": this seems not so surprising, because we already know that diffusion models can synthesize novel images (like flying monkeys) not existed before.
>
> In unconditional diffusion modeling set up, they can synthesize non-existing faces that looks convincing. In conditional modelling set up (e.g. text2image), they can synthesize image corresponding to prompt combinations that are unlikely to occur in the corpus.
>
> Indeed, we know diffusion models can synthesize non existing natural looking images (e.g. unconditional novel face generation, conditional text 2 novel image generation), but to synthesize novel samples that are consistent with certain “underlying rules”, it’s not known yet. For natural images, such “rules” kind of exist in the form of natural statistics of object and patterns, but it’s hard to evaluate whether the samples are “rule” consistent or not. For single sample evaluation, people usually resort to Visual Question Answering (VQA) by human or machine for rough evaluations of sample quality. Our discrete setting allows precise evaluation of samples about the extent to which they are rule consistent.
>
> The other surprising aspect arise from the contrast to the theory of diffusion model: if we optimize the denoising score matching loss to the global minimum with arbitrary score function approximators, it will result in the memorization of training data, i.e. delta measure over the data points, so no novel sample will be generated.
>
> So our result is significant regarding the generalization of diffusion models: i.e. they not just generalize but also do so “intelligently” and obeys the unobserved rules of data generation, which pushes the “limit” of what they could generalize to and how.
>
> > "memorized and recombined local parts of the training samples": **this is an interesting point, because previously we did not know how diffusion models can learn the structure.** However, in this paper, they have this claim mainly because of Figure 7, which is a plot showing the model first learns local part, then global parts. It would be better if the authors can provide deeper discussions on this point.
> >
>
> We agree with the reviewer. We observed and validated this intriguing phenomenon in our setup. We have also been searching for an deeper understanding of it, which will be part of our future work.
>
> > "advanced sampling techniques were needed for inpainting": this is also an interesting point, but it seems that here the authors simply mean that Sequential Monte Carlo will give better performance, but did not provide any deep analysis.
> >
>
> We have performed ablation analysis for the hyper parameters for the Sequential Monte Carlo method (i.e. Twisted Diffusion Sampler) as in Fig. 9 in the supplementary of the paper. Basically we can see there is some gain in adding to the population size of particles and there is more gain in adding to the number of steps of the sampler. Generally, since the TDS was asymptotically exact, a larger step number and population size should result in a better approximation of the conditional probability, which lead to better panel completion accuracy. In contrast, Repaint or DDNM methods are heuristic method, so they may not enjoy the same properties as TDS did [^1].
>
> [^1] Conditional sampling within generative diffusion models https://arxiv.org/abs/2409.09650v1
>
> > "pattern completion capability can generalize to rules unseen": this is an interesting point, but it seems that the authors mainly provide some unseen examples in test set, and show that diffusion models can work well on these examples. The fact that models can generalize to unseen cases has been observed by many previous studies, **and it would be great if the authors can use Raven's task to provide better intuitions and explanations.**
>
> Generally, we think it could be framed as a form of “in context learning” of diffusion models, which is usually described in term of “inferring” the latent rule and sampling missing part accordingly. The exact circuit mechanism of which is currently still not well understood, and could be better dissected in future work.

---

> > ### Author Response · Authors · 2024-11-25
> > **Responses to the weaknesses of the paper and new experimental results regarding the generalization of rule representation [Cont'd]**
> >
> > > "rule representation": this is not in particular surprising, because people previously already observe that deep networks can learn semantic features, or self-supervised learning framework can learn very strong features.
> > >
> >
> > Yeah indeed, as we have mentioned in the paper, in the natural image domain, people have found diffusion models can learn object classification features via unconditional generative modeling task. Our current result is one step further, showing diffusion models can also learn strong features for abstract rules via generative modeling, not just visual semantic categories.
> >
> > The more surprising aspect of this result is that we show the representation is very **generalizable**, even to rules unseen during diffusion training.
> > Per the request of the reviewer `Cqnf`, we have conducted a new set of experiments focusing on the generalization of representation. We trained unconditional diffusion (DiT-S) with more held out rules from the 40 rules and then use the same linear probe procedure to classify rules from the emerged hidden representation.
> > We found, that even with 23 rules held out during generative training, the emerged representation can still linearly classify 40 rules at 98% test accuracy (seen rules, 99%, unseen rules 96%). Full results are shown in following Tab. 1. This indicates that, even without seeing many rules, the diffusion model can learn general representations for all the rules in that domain.
> >
> > Tab 1. **Rule classification accuracy of diffusion models trained with increasing fraction of held out rules**.
> > ***Test Acc, Train Acc***: the train and test accuracy of linear probe (averaged across 40 rules) at 1M steps, at block.11, average token representation, t=25.0 .
> > ***Test Acc*, Train Acc*, step****: the best test accuracy achieved through training (at step*) and their corresponding train accuracy. All best test accuracy happened to be achieved at final layer (block.11), and with avg token representation.
> >
> > | ExpNote | heldout_num | heldout_rules | Test Acc | Train Acc | Test Acc* | Train Acc* | step* |
> > | --- | --- | --- | --- | --- | --- | --- | --- |
> > | BalSetx1 | 5 | (1, 16, 20, 34, 37) | 0.997 | 1.000 | 0.997 | 1.000 | 1000000 |
> > | BalSetx2 | 10 | (1, 8, 12, 16, 20, 24, 34, 36, 37, 39) | 0.997 | 1.000 | 0.997 | 1.000 | 1000000 |
> > | BalSetx3 | 15 | (1, 5, 8, 12, 16, 17, 20, 21, 24, 33, 34, 36, 37, 38, 39) | 0.991 | 1.000 | 0.991 | 1.000 | 1000000 |
> > | BalSetx4 | 19 | (1, 3, 5, 8, 10, 12, 16, 17, 20, 21, 24, 29, 31, 33, 34, 36, 37, 38, 39) | 0.986 | 1.000 | 0.991 | 1.000 | 700000 |
> > | BalSetx5 | 23 | (0, 1, 3, 5, 8, 10, 12, 14, 16, 17, 20, 21, 24, 27, 29, 31, 33, 34, 35, 36, 37, 38, 39) | 0.976 | 0.995 | 0.986 | 0.998 | 500000 |
> > | BalSetx6 | 27 | (0, 1, 3, 4, 5, 8, 10, 12, 14, 16, 17, 19, 20, 21, 24, 26, 27, 29, 30, 31, 33, 34, 35, 36, 37, 38, 39) | 0.880 | 0.922 | 0.976 | 0.994 | 500000 |
> > | Attr0 | 10 | (0, 1, 2, 3, 4, 5, 6, 7, 8, 9) | 0.978 | 0.988 | 0.979 | 0.987 | 700000 |
> > | Attr3 | 10 | (30, 31, 32, 33, 34, 35, 36, 37, 38, 39) | 0.991 | 0.999 | 0.994 | 1.000 | 700000 |
> > | Rel0 | 4 | (0, 10, 20, 30) | 0.996 | 1.000 | 0.996 | 1.000 | 1000000 |
> > | Rel3 | 4 | (3, 13, 23, 33) | 0.998 | 1.000 | 0.998 | 1.000 | 1000000 |
> > | Rel5 | 4 | (5, 15, 25, 35) | 0.996 | 1.000 | 0.996 | 1.000 | 700000 |
> > | Rel8 | 4 | (8, 18, 28, 38) | 0.994 | 0.999 | 0.994 | 0.999 | 1000000 |
> > | Rel012 | 12 | (0, 1, 2, 10, 11, 12, 20, 21, 22, 30, 31, 32) | 0.997 | 1.000 | 0.997 | 1.000 | 1000000 |
> > | Rel014 | 12 | (0, 1, 4, 10, 11, 14, 20, 21, 24, 30, 31, 34) | 0.998 | 1.000 | 0.998 | 1.000 | 1000000 |
> > | Rel023 | 12 | (0, 2, 3, 10, 12, 13, 20, 22, 23, 30, 32, 33) | 0.996 | 1.000 | 0.996 | 1.000 | 700000 |
> > | Rel034 | 12 | (0, 3, 4, 10, 13, 14, 20, 23, 24, 30, 33, 34) | 0.997 | 1.000 | 0.997 | 1.000 | 1000000 |
> > | Rel123 | 12 | (1, 2, 3, 11, 12, 13, 21, 22, 23, 31, 32, 33) | 0.997 | 1.000 | 0.997 | 1.000 | 1000000 |
> > | Rel234 | 12 | (2, 3, 4, 12, 13, 14, 22, 23, 24, 32, 33, 34) | 0.996 | 1.000 | 0.996 | 1.000 | 1000000 |
> > | Rel56 | 8 | (5, 6, 15, 16, 25, 26, 35, 36) | 0.992 | 1.000 | 0.992 | 1.000 | 1000000 |
> > | Rel01234 | 20 | (0, 1, 2, 3, 4, 10, 11, 12, 13, 14, 20, 21, 22, 23, 24, 30, 31, 32, 33, 34) | 0.991 | 1.000 | 0.993 | 1.000 | 700000 |
> > | Rel789 | 12 | (7, 8, 9, 17, 18, 19, 27, 28, 29, 37, 38, 39) | 0.950 | 0.968 | 0.954 | 0.970 | 700000 |
> > | Rel1234 | 16 | (1, 2, 3, 4, 11, 12, 13, 14, 21, 22, 23, 24, 31, 32, 33, 34) | 0.997 | 1.000 | 0.997 | 1.000 | 700000 |
> > | Rel0123456 | 28 | (0, 1, 2, 3, 4, 5, 6, 10, 11, 12, 13, 14, 15, 16, 20, 21, 22, 23, 24, 25, 26, 30, 31, 32, 33, 34, 35, 36) | 0.766 | 0.808 | 0.963 | 0.991 | 200000 |

---

### Official Review · Reviewer_nFii · 2024-10-27

**Soundness:** 2
**Presentation:** 2
**Contribution:** 2
**Rating:** 3
**Confidence:** 4

**Summary:**

In this study, the authors assess the capabilities of existing diffusion and language models in the context of abstract reasoning. They represent Raven's Progressive Matrices (RPM) puzzles in a numeric format, mapping each visual attribute to an integer value. The authors conduct several ablation studies on the trained models to evaluate their abstract reasoning abilities.

**Strengths:**

In this work, the following strengths can be found:
1. They conduct experiments on various state-of-the-art diffusion models as well as GPT-2. These models are trained on a proposed simplified version of Raven's Progressive Matrices, which is suitable for models that lack visual modalities, such as GPT-2.
2. In addition to training existing models on the proposed dataset, the authors provide extensive evaluations and ablation studies to explore the advantages of diffusion-based models compared to language models or better sequence generators, like GPT-2, in this context.

**Weaknesses:**

Some of the weaknesses of this work are:

### The submitted document needs to be completed.
1. The paper is incomplete. The supplementary material includes sections that are missing content (B.1, B.3, B.4). Some sections are only partially completed, such as B.2.3, which is missing figures and explanations that cannot be linked to experimental results, along with 'XXXXX' entries. Additionally, in section A.2.2, line 748, the authors failed to provide the hyperparameter values. Although these issues are present in the supplementary material, the reviewer believes that the authors should have been more diligent and submitted a complete version of their work.


### Clarity and Presentation
The reviewer suggests that the authors revisit their work, as improvements can be made in the presentation of the paper. The figures should be organized more effectively; currently, Figures 2, 3, and others appear to be placeholders with individual images stacked. Additionally, there are other issues with the presentation of the results:
1. In Figure 5.D, not all versions are visible. The label "scale" is unclear in the legend, and it seems different models are referenced. What does "model_class" refer to?
2. In Fig. 2.D, what does the individual held-out rule color correspond to? There are five different cases, but the authors do not mention what each of them represents.

### Missing References to Prior Work
Previous research has explored ways to represent Raven's Progressive Matrices (RPMs) in a format suitable for language models. The reviewer suggests that the authors should reference these studies and compare their RPM representation with existing works. It is possible that the proposed representation may not be appropriate for language models. Here are some relevant references: [1], [2], [3]. Additionally, this work introduces a simplified RPM dataset [4], which appears to be quite similar to the proposed representation.

### Dataset Concerns
1. The reviewer expresses skepticism regarding the validity and difficulty level of the tested out-of-distribution (OOD) and held-out scenarios. In the proposed simplified dataset, all data is numeric. Therefore, if a model is trained to learn the constancy relation between the numbers on the second channel of the RPM, it may not be very challenging to identify this type of relationship on the first channel. This scenario differs from visual puzzles, where constancy in color is present during training, but testing introduces puzzles with constancy in shape for the first time.


### Contribution Concerns
1. The contribution of this work seems to be limited. The reasons are that the authors do not seem to propose a novel component but rather re-train existing models for a different reasoning dataset.

[1] Zhang, Chengru, and Liuyun Wang. "Evaluating Abstract Reasoning and Problem-Solving Abilities of Large Language Models Using Raven's Progressive Matrices." (2024).

[2] Webb, Taylor, Keith J. Holyoak, and Hongjing Lu. "Emergent analogical reasoning in large language models." Nature Human Behaviour 7.9 (2023): 1526-1541.

[3] Ahrabian, K., Sourati, Z., Sun, K., Zhang, J., Jiang, Y., Morstatter, F., & Pujara, J. (2024). The curious case of nonverbal abstract reasoning with multi-modal large language models. arXiv preprint arXiv:2401.12117.

[4] Schug, S., Kobayashi, S., Akram, Y., Sacramento, J., & Pascanu, R. (2024). Attention as a Hypernetwork. arXiv preprint arXiv:2406.05816.

**Questions:**

1. For the ablations, the authors seem to use it as their reference model. However, SiT seems to perform better than DiT. Why did the authors not include SiT in the performed ablations?
2. How do the authors ensure that the training and testing set puzzles do not overlap? The dataset seems huge: 1,200,000 row samples per rule, with 40 rules in total.

---

### Official Review · Reviewer_QARN · 2024-10-31

**Soundness:** 3
**Presentation:** 1
**Contribution:** 2
**Rating:** 3
**Confidence:** 3

**Summary:**

This work studies the ability of diffusion models to learn abstract rules and apply them during the generation of missing parts. Several families of diffusion models with different backbones (UNet and Diffusion Transformer) are trained and tested on the Raven's Progressive Matrices, a well-known reasoning task.
The authors show that in the unconditional generation task, diffusion models are able to generate consistent samples, which demonstrates their success in learning hidden rules. In the conditional generation task where the objective is to generate a missing panel, it is noted that the accuracy of the completion critically depends on the sampling algorithm. In addition, the authors illustrate that the diffusion models have some internal representations that distinguish the learned rules, evidenced by the high classification accuracy of a linear model trained on the hidden states of the diffusion models. This work provides a detailed discussion on the observed behaviour of diffusion models during training and testing, concluding that diffusion models are capable of performing rule-learning given a large data scale.

**Strengths:**

- This work is one of the first to evaluate a wide range of diffusion models' capability in rule-learning, targeting the task of Raven's Progressive Matrices.
- Problem description and formulation is well illustrated in the overview diagram.
- Most of the claims are supported by solid results.
- This work provides detailed insights on diffusion models' behaviour in the task of rule-learning and reasoning, discussing potential explanations of those behaviours.

**Weaknesses:**

- Formal writing: the authors should carefully check all typos and grammar issues in this paper. Examples include but are not limited to: incorrect openings of quotation marks, incomplete sentences (Section 3.3, around line 180), informal use of abbreviations ("esp."), inconsistent naming (Raven's progressive matrix and Raven's progression matrix).
- Organization and presentation of results: this work includes several crowded figures, when they are referenced in the main text, it is often hard to follow since the text and the reference figure are far away from each other. For example, Section 4.3 involves discussions on Figure 4, Figure 10 and Figure 11, where Figure 10 and 11 are only found in the appendix. These results can potentially be summarized into one or two high-level figures that present the major outcomes, with detailed diagrams or tables included in the appendix. This would help readers locate relevant information easily.
- The claim on comparisons to auto-regressive models: the auto-regressive models used for comparison are GPT2 models, which might be out-dated given that there exist much stronger models. Out-performing GPT2 models does not support the claims on out-performing auto-regressive models.

**Questions:**

- How is GenRAVEN different from other RPM datasets in the literature? Is the encoding introduced in this work a novel transformation of RPM images to numerical data?
- In Section 4.1, it is mentioned that Logic-Attribute rules are hard to learn and will be discussed in Section 6, where there is only a brief observation on the correlation between panel overlapping rate and the performance per rule type. Can you elaborate more on why does that make the Logic-Attribute type different from other types?
- As mentioned in weaknesses, is it possible to include evaluations of more recent auto-regressive models for comparisons? For example, Llama 3.

---

### Official Review · Reviewer_PNoL · 2024-11-01

**Soundness:** 3
**Presentation:** 3
**Contribution:** 2
**Rating:** 5
**Confidence:** 4

**Summary:**

This paper evaluates the abilities of diffusion models in contrast to autoregressive models on the problem of Raven’s progression matrix. It proposes a dataset based on 40 relational rules and performs various types of experiments to evaluate the learning procedures and abilities of the trained models. It concludes that diffusion models has certain advantages over autoregressive models while in “rule consistent panel completion”, diffusion models happen to be weaker.

**Strengths:**

Paper is well written.

Experiments on the RPM are detailed and insightful.

It proposes a new dataset that may be useful for the community.

**Weaknesses:**

Literature review misses relevant prior work.

[1] Lachapelle, S., Mahajan, D., Mitliagkas, I. and Lacoste-Julien, S., 2023. Additive decoders for latent variables identification and cartesian-product extrapolation. Advances in Neural Information Processing Systems, 36.

[2] Bansal, A., Schwarzschild, A., Borgnia, E., Emam, Z., Huang, F., Goldblum, M. and Goldstein, T., 2022. End-to-end algorithm synthesis with recurrent networks: Extrapolation without overthinking. Advances in Neural Information Processing Systems, 35, pp.20232-20242.


Both of these papers consider a setting that has certain similarities to the problem that this paper studies. [1] also considers attributes such as location and color.



-----


The experiments of the paper are rather limited as it only evaluates the diffusion models on one specific dataset that is proposed for the first time in the paper. The dataset itself is not large (40 relational rules and 3x3 matrix of panels), and the achieved accuracy is rather high (99.8%) indicating that it is not very challenging. In my view, to draw conclusions about the learning abilities and learning procedures of diffusion models one would need to see more convincing experimental results on a broader set of related tasks.


------

While the paper contrasts the abilities of diffusion models against that of autoregressive models, and provides some useful empirical evidence about the cases where one performs better than the other, it is hard to be sure that these would generalize beyond the specific task of Raven’s progression matrix. In its current form, I think it would be more appropriate for the paper to change its title and conclusions to reflect that it is only about the Raven’s progression matrix and not the broader topic of abstract reasoning. But, this makes the scope of the paper rather limited. So, I would suggest that the paper expands its experiments to other reasoning tasks, and try to make broader conclusions beyond the RPM task, perhaps for a certain class of reasoning tasks.

**Questions:**

Please see weaknesses. Would love to hear any clarifications or additional experiments that authors might want to share.

If authors manage to extend their experiments to some existing reasoning tasks in the literature, for example, one of the papers mentioned above (or any other existing reasoning task that they find appropriate), it would make their results more convincing and more connected to the existing literature.

---

### Official Review · Reviewer_CZnK · 2024-11-02

**Soundness:** 2
**Presentation:** 3
**Contribution:** 3
**Rating:** 6
**Confidence:** 4

**Summary:**

The paper investigates how diffusion models learn abstract rules using synthetic data from the Raven's Progressive Matrices (RPM) task, offering a detailed comparison with autoregressive models. The core contributions are as follows:

1. Demonstrating that diffusion models exhibit significantly less memorization than autoregressive models when trained on RPM data.
2. Showing that diffusion models generate outputs hierarchically, learning to recombine elements at a local scale before forming broader structures.
3. Revealing that the intermediate representations within diffusion models align closely with the underlying rules of the provided samples.
4. Observing that diffusion models only start to effectively learn abstract rules when a certain data scale is reached.

**Strengths:**

1. Comprehensive and well-designed experiments, particularly the use of synthetic RPM data, which introduces a new perspective on studying diffusion models. This controlled approach contrasts with natural object generation, which often lacks experimental control, allowing a more precise investigation of rule learning.

2. The experiments are rigorous, with careful comparisons between diffusion and autoregressive models, examining factors like memorization and hierarchical generation. This provides strong support for the paper’s findings.

3. The paper is well-organized, presenting methods and results in clearly.

This work broadens our understanding of diffusion models, suggesting their applicability beyond visual generation to reasoning tasks, and providing a framework for controlled abstract rule learning studies.

**Weaknesses:**

Object Tokenization: The study focuses solely on the learning of abstract rules, assuming idealized perception of object features. While this approach isolates rule learning effectively, the paper could benefit from a comparison between pixel tokenization and object/panel tokenization, showing how different tokenization strategies might impact results.

Autoregressive Model Setup: In Section A.3, the authors apply object tokenization in autoregressive modeling by simply combining object features. This factorized approach assumes independence among features within the same object, differing from the diffusion model’s approach which models the joint strictly. It would strengthen the study to model the joint probability of object attributes directly.

RPM for Transfer Learning: The Raven's Progressive Matrices (RPM) test is typically a transfer learning process for humans, where test-takers encounter the task without prior exposure or practice. In contrast, diffusion models in this study train on RPM with extensive sample exposure. It would be valuable to explore how diffusion models trained on natural images could transfer to this task and perform zero-shot rule discovery, likely revealing contrasts with the training approach described in the paper.

Contribution Framing: Some contributions could be framed more effectively. For instance, “showing that unconditional diffusion models can be used for conditional inference, i.e., completing missing panels consistent with the rule” may be perceived as a weaker contribution since converting unconditional diffusion models for inpainting is already well-known. A revised framing of contributions could clarify the study’s unique insights.

**Questions:**

I was surprised to see that autoregressive models performed worse than diffusion models. In human visual reasoning, saccadic movements often create a dynamic, sequential prediction process resembling autoregressive models. Could the authors provide some intuition as to why autoregressive models underperform in this context? It would be interesting to understand potential limitations or factors influencing these results within the framework of this paper.

---

### Official Review · Reviewer_Cqnf · 2024-11-04

**Soundness:** 2
**Presentation:** 2
**Contribution:** 2
**Rating:** 5
**Confidence:** 4

**Summary:**

The paper presents a study of text-free diffusion models performance on a task derived from the Ravens Matrices "GeNRaven", using 5 held out rules as well as per-rule training and test sets to assess whether the diffusion model can

1. unconditionally generate valid rows following a single rule (C3)
2. conditionally infill a valid square when presented a ravens matrix.

Authors find that C3 performance generally exceeds infill performance (except for held out rules) and that linear probes can extract classifiers of _all_ rules (including held out rules) from latent representations. The performance is highly sensitive to training parameters and sampling methods used.

**Strengths:**

- the papers approach is very original and well designed (good comparisons, good ablations, mainly well done experiments, most relevant literature cited)
- quality-clarity: exposition is well done, quality of the paper is good except for the weaknesses below
- significance: I think the questions raised with the preset experiments and potential followups make this a meaningful benchmark to understand compositional generalization in diffusion models

**Weaknesses:**

- I don't think reasoning is the appropriate term to use, instead, the authors have found a wonderful way to study _compositional generalization_ https://arxiv.org/abs/2307.05596 this has been found to allow diffusion models to even _interpolate_  latent variables https://arxiv.org/abs/2405.19201 and in the specific holdout setting the authors studied (leave-one-out holdout rules), everything is explained by compositional generalization (the semantic debate whether or not this is "true reasoning" is uninteresting to me here). Reasoning  might include comp. generalization, but also includes things like multi planning (see e.g. https://www.arxiv.org/abs/2409.13373) so the paper should change its framing of what is being studied
- the paper should repeat the strongest results with increasing subsets of held out rule to see whether the breakdown follows the expectations set in the current interpretation as a sanity check -science is about "kill your babies". this is in particular interesting for the linear probe (how many times must a factor appeat to be visible?)
- I am unsure about the sample sizes chosen being meaningful, please use something like https://cran.r-project.org/web/packages/effectsize/index.html to justify experimental design. where appropriate, statistical tests should be used to justify claims against a concrete null hypothesis
- was only a single seed run for everything (in particular, training the model)? if yes, please rerun/retrain, do a statistical significance test and report uncertainties (I only see this on evaluation right now)

**Questions:**

1. What happens if you condition on completely unseen shapes, bust consistent rules or vice versa?
2. can you factorize the readout with a dimensionality constraint and retain the performance?
3. How do you perform scoring ? manually or an automated evaluation pipeline?

---

> ### Author Response · Authors · 2024-11-18
> **Thanks for the review. Results from New Training Experiment addressing increasing heldout rules and reproducibility across runs [Part1]**
>
> > I don't think reasoning is the appropriate term to use, instead, the authors have found a wonderful way to study *compositional generalization* https://arxiv.org/abs/2307.05596 ……
> >
>
> We agree with the reviewer, that our task is indeed a nice setting to study compositional generalization. Specifically, the unconditional and conditional generation performance on held-out rules can be well framed as a test for compositional generalization, e.g. constant shape + progression size ⇒ constant size. Indeed we observed intriguing non-monotonic learning dynamics for these held-out rules.
>
> On the other hand, we think the conditional generation / panel completion experiment can be framed as reasoning, since it potentially involves inferring the underlying rules from observed samples and applying it to the occluded row. (which is also the underlying assumption for the corresponding task in human.)
> One could argue that the sampling steps in the diffusion or autoregressive model could be analogous to reasoning steps. Further previous works also frame RAVEN's task as abstract visual reasoning [^1][^2], so we think it’s justified to call the task reasoning task.
>
> [1] Chi Zhang*, Feng Gao*, Baoxiong Jia, Yixin Zhu, Song-Chun Zhu [RAVEN: A Dataset for Relational and Analogical Visual rEasoNing](http://wellyzhang.github.io/attach/cvpr19zhang.pdf)
>
> [2] Małkiński M, Mańdziuk J. Deep learning methods for abstract visual reasoning: A survey on raven's progressive matrices. arXiv preprint arXiv:2201.12382. 2022 Jan 28.
>
> > the paper should repeat **the strongest results with increasing subsets of held out rule to see whether the breakdown follows the expectations set in the current interpretation as a sanity check -science is about "kill your babies". this is in particular interesting for the linear probe (how many times must a factor appeat to be visible?**)
> >
>
> We totally agree with the reviewer, and we conducted two sets of new experiments, and trained 24 new DiT models to address this with increasingly many held-out rules.
>
> In the first set of experiments, we held out an increasing number of rules from 5, 10, 15, 19, 23, 27 rules. Given the combinatorial nature of selecting held out rules, we cannot exhaust all combinations, so we tried our best to balance the number of held out rules within each rule types at each held out number. We call them balanced set held out (`BalSetx1` - `BalSetx6`)
> In the second set of experiments, we held out whole row or column in the rule matrix, either all the relations for one attributes (shape, number / pos, 10 rules held out, denoted as `attr0`- `attr3`) or all the attributes for one or several relations (denoted as `rel0`, `rel01234` … 4,8,12,..28 rules held out). The specifics of held out rules and the total number are listed in the attached Table 1.
>
> Specifically, we trained unconditional diffusion model (DiT-S/1) on the remaining rules with 4000 RPMs per rule. Then we used the same protocol to fit linear probes for 40 rules to evaluate rule classifying representations emerged in the model.
>
> To our delight, the diffusion model’s learned representations are very robust to holding out rules during training: even with 23 held-out rules (`BalSetx5`) the final layer representation after training has test accuracy 97.6%, during training the best test accuracy was achieved at 0.5M steps at 98.6%.
> Further, we also found non-monotonic learning dynamics for these representations, i.e. when there are too many held-out rules, the latent representation will initially be able to accurately classify rules, while over-training will lead to the deterioration of representations.
> With 27 heldout rules, the representation quality broke down quit a bit: test accuracy peaked at 97.6% at 0.5M step, and dereased to 88% after 1M step training.
>
> All in all, these additional experiments further showed that diffusion models are capable of constructing strong generalizing representation for underlying factor (e.g. rules), even when only have seen a limited portion of data.

---

> > ### Author Response · Authors · 2024-11-18
> > **Thanks for the review. Results from New Training Experiment addressing increasing heldout rules and reproducibility across runs [Part2]**
> >
> > Tab 1. **Rule classification accuracy of diffusion models trained with increasing fraction of held out rules**.
> > ***Test Acc, Train Acc***: the train and test accuracy of linear probe (averaged across 40 rules) at 1M steps, at block.11, average token representation, t=25.0 .
> > ***Test Acc*, Train Acc*, step****: the best test accuracy achieved through training (at step*) and their corresponding train accuracy. All best test accuracy happened to be achieved at final layer (block.11), and with avg token representation.
> >
> > | ExpNote | heldout_num | heldout_rules | Test Acc | Train Acc | Test Acc* | Train Acc* | step* |
> > | --- | --- | --- | --- | --- | --- | --- | --- |
> > | BalSetx1 | 5 | (1, 16, 20, 34, 37) | 0.997 | 1.000 | 0.997 | 1.000 | 1000000 |
> > | BalSetx2 | 10 | (1, 8, 12, 16, 20, 24, 34, 36, 37, 39) | 0.997 | 1.000 | 0.997 | 1.000 | 1000000 |
> > | BalSetx3 | 15 | (1, 5, 8, 12, 16, 17, 20, 21, 24, 33, 34, 36, 37, 38, 39) | 0.991 | 1.000 | 0.991 | 1.000 | 1000000 |
> > | BalSetx4 | 19 | (1, 3, 5, 8, 10, 12, 16, 17, 20, 21, 24, 29, 31, 33, 34, 36, 37, 38, 39) | 0.986 | 1.000 | 0.991 | 1.000 | 700000 |
> > | BalSetx5 | 23 | (0, 1, 3, 5, 8, 10, 12, 14, 16, 17, 20, 21, 24, 27, 29, 31, 33, 34, 35, 36, 37, 38, 39) | 0.976 | 0.995 | 0.986 | 0.998 | 500000 |
> > | BalSetx6 | 27 | (0, 1, 3, 4, 5, 8, 10, 12, 14, 16, 17, 19, 20, 21, 24, 26, 27, 29, 30, 31, 33, 34, 35, 36, 37, 38, 39) | 0.880 | 0.922 | 0.976 | 0.994 | 500000 |
> > | Attr0 | 10 | (0, 1, 2, 3, 4, 5, 6, 7, 8, 9) | 0.978 | 0.988 | 0.979 | 0.987 | 700000 |
> > | Attr3 | 10 | (30, 31, 32, 33, 34, 35, 36, 37, 38, 39) | 0.991 | 0.999 | 0.994 | 1.000 | 700000 |
> > | Rel0 | 4 | (0, 10, 20, 30) | 0.996 | 1.000 | 0.996 | 1.000 | 1000000 |
> > | Rel3 | 4 | (3, 13, 23, 33) | 0.998 | 1.000 | 0.998 | 1.000 | 1000000 |
> > | Rel5 | 4 | (5, 15, 25, 35) | 0.996 | 1.000 | 0.996 | 1.000 | 700000 |
> > | Rel8 | 4 | (8, 18, 28, 38) | 0.994 | 0.999 | 0.994 | 0.999 | 1000000 |
> > | Rel012 | 12 | (0, 1, 2, 10, 11, 12, 20, 21, 22, 30, 31, 32) | 0.997 | 1.000 | 0.997 | 1.000 | 1000000 |
> > | Rel014 | 12 | (0, 1, 4, 10, 11, 14, 20, 21, 24, 30, 31, 34) | 0.998 | 1.000 | 0.998 | 1.000 | 1000000 |
> > | Rel023 | 12 | (0, 2, 3, 10, 12, 13, 20, 22, 23, 30, 32, 33) | 0.996 | 1.000 | 0.996 | 1.000 | 700000 |
> > | Rel034 | 12 | (0, 3, 4, 10, 13, 14, 20, 23, 24, 30, 33, 34) | 0.997 | 1.000 | 0.997 | 1.000 | 1000000 |
> > | Rel123 | 12 | (1, 2, 3, 11, 12, 13, 21, 22, 23, 31, 32, 33) | 0.997 | 1.000 | 0.997 | 1.000 | 1000000 |
> > | Rel234 | 12 | (2, 3, 4, 12, 13, 14, 22, 23, 24, 32, 33, 34) | 0.996 | 1.000 | 0.996 | 1.000 | 1000000 |
> > | Rel56 | 8 | (5, 6, 15, 16, 25, 26, 35, 36) | 0.992 | 1.000 | 0.992 | 1.000 | 1000000 |
> > | Rel01234 | 20 | (0, 1, 2, 3, 4, 10, 11, 12, 13, 14, 20, 21, 22, 23, 24, 30, 31, 32, 33, 34) | 0.991 | 1.000 | 0.993 | 1.000 | 700000 |
> > | Rel789 | 12 | (7, 8, 9, 17, 18, 19, 27, 28, 29, 37, 38, 39) | 0.950 | 0.968 | 0.954 | 0.970 | 700000 |
> > | Rel1234 | 16 | (1, 2, 3, 4, 11, 12, 13, 14, 21, 22, 23, 24, 31, 32, 33, 34) | 0.997 | 1.000 | 0.997 | 1.000 | 700000 |
> > | Rel0123456 | 28 | (0, 1, 2, 3, 4, 5, 6, 10, 11, 12, 13, 14, 15, 16, 20, 21, 22, 23, 24, 25, 26, 30, 31, 32, 33, 34, 35, 36) | 0.766 | 0.808 | 0.963 | 0.991 | 200000 |
> >
> > > I am unsure about the sample sizes chosen being meaningful, please use something like https://cran.r-project.org/web/packages/effectsize/index.html to justify experimental design. Where appropriate, statistical tests should be used to justify claims against a concrete null hypothesis
> > >
> >
> > We totally agree with reviewer, that we shall improve the statistical rigor in machine learning research. Since there are many analysis and results through the paper, is there a particular claim or test that you think we should focus on? (See also the response to next concern.)

---

> > > ### Author Response · Authors · 2024-11-18
> > > **Thanks for the review. Results from New Training Experiment addressing increasing heldout rules and reproducibility across runs [Part3]**
> > >
> > > > was only a single seed run for everything (in particular, training the model)? if yes, please rerun/retrain, do a statistical significance test and report uncertainties (I only see this on evaluation right now)
> > >
> > > Thanks for the suggestion for statistical testing. In the paper, we majorly reported results on one run, because we have conducted pilot experiments which shows the training of diffusion model is fairly consistent and reproducible across initial seeds of optimization.
> > > This is consistent with others’ previous finding in literature [^3]. Note that this reproducibility is even stronger than the statistical level, but actually it’s at sample level! Different diffusion models trained on the same / similar data will learn almost the same mapping from initial noise to sample with deterministic sampler (which is what we majorly used).
> > > Because of this, we infer, the uncertainty of evaluation metric is majorly attributable to the uncertainty in sampling. Namely, C3, valid row, or memorization fraction are all estimation of probability through fraction in a finite sample size. So they will naturally have uncertainty estimates associated with fractions (i.e. beta distributions).
> > > We have added the confidence interval of C3 and valid rows in our main table also as following.
> > >
> > > It would be infeasible to re-run all the training experiments. (On a H100, DiT-S would take 20hrs to train, and DiT-B would take 40 hrs to train, and the computational resource is currently limited.)
> > > However, to prove our point, upon reviewer’s request, we re-run the training of DiT-S and DiT-B models three times with the same configuration on same data (5 heldout rules), with different random seeds. We examined the previously mentioned reproducibility on the sample level. i.e. using the ***same noise seed to generate the initial noise*** image, and then sample through the two diffusion models trained separately with different random seeds during training, using DDIM with 100 steps.
> > >
> > > At DiT-S model scale, within 10240 samples, DiT-S model 1 has 6505 C3 samples (63.5%, 95% CI [62.6%, 64.5%]), while DiT-S model 2 has 6564 C3 samples (64.1%, 95% CI [63.2%, 65.0%]). We can see the C3 ratio estimated for the two models fall into the confidence interval of the other one. Further, this consistency and reproducibility is stronger than the marginal statistical level. When we examined the 10240 samples from model 1 and model 2 with the matching initial noise: 8309 samples (81%) has the same C3 rule (or none). 22037 out of 30720 rows (72%) have the same rule set applied. 73% of the sample attribute entries have the same value. Though these ratios are not 100%, it shows that even separately trained Diffusion model on the same data will have largely reproducible samples when using the same noise seed.
> > >
> > > Further, at DiT-B model scale, the 10240 samples generated by the three DiT-B models are exactly the same at the **attribute** **entry level and every level**, showing full sample reproducibility. This fully justifies that for larger scale diffusion model (≥ DiT-B) the we do not need to train multiple runs, since they will yield the same sample after training. Thus, we should focus more on the uncertainty associated with noise sampling instead of training.
> > >
> > > **Table 2. Statistical consistency of performance of DiT models across repetition in training. 10240 total samples, with the same initial noise for all 6 models.**
> > >
> > > |  | C3 count | C3 frac | C3 ci L | C3 ci U | valid count | valid frac | valid ci L | valid ci U |
> > > | --- | --- | --- | --- | --- | --- | --- | --- | --- |
> > > | DiT-S_rep1 | 6627 | 0.647 | 0.638 | 0.656 | 24229 | 0.789 | 0.784 | 0.793 |
> > > | DiT-S_rep2 | 6556 | 0.640 | 0.631 | 0.649 | 23925 | 0.779 | 0.774 | 0.783 |
> > > | DiT-S_rep3 | 6629 | 0.647 | 0.638 | 0.657 | 24107 | 0.785 | 0.780 | 0.789 |
> > > | DiT-B_rep1 | 6542 | 0.639 | 0.630 | 0.648 | 24143 | 0.786 | 0.781 | 0.790 |
> > > | DiT-B_rep2 | 6542 | 0.639 | 0.630 | 0.648 | 24143 | 0.786 | 0.781 | 0.790 |
> > > | DiT-B_rep3 | 6542 | 0.639 | 0.630 | 0.648 | 24143 | 0.786 | 0.781 | 0.790 |
> > >
> > > [^3] Zhang H, Zhou J, Lu Y, Guo M, Wang P, Shen L, Qu Q. The emergence of reproducibility and consistency in diffusion models. InForty-first International Conference on Machine Learning 2023.

---

> > > > ### Comment · Reviewer_Cqnf · 2024-11-25
> > > >
> > > > thank you for providing strong evidence for omitting the per-seed level, as well as justifying the use of  single seed

---

> > > ### Comment · Reviewer_Cqnf · 2024-11-25
> > >
> > > > We totally agree with reviewer, that we shall improve the statistical rigor in machine learning research. Since there are many analysis and results through the paper, is there a particular claim or test that you think we should focus on? (See also the response to next concern.)
> > >
> > > the easy answer would be "all of them", but the most important one I find the central claim: how many rules must there be  in total and held out for an apparent generalization to be unlikely enough to be implausible (I think you have the sample size sufficient for this based on other results, but formalizing the claim would be helpful imo)

---

> > ### Comment · Reviewer_Cqnf · 2024-11-25
> >
> > >To our delight, the diffusion model’s learned representations are very robust to holding out rules during training: even with 23 held-out rules (BalSetx5) the final layer representation after training has test accuracy 97.6%, during training the best test accuracy was achieved at 0.5M steps at 98.6%. Further, we also found non-monotonic learning dynamics for these representations, i.e. when there are too many held-out rules, the latent representation will initially be able to accurately classify rules, while over-training will lead to the deterioration of representations. With 27 heldout rules, the representation quality broke down quit a bit: test accuracy peaked at 97.6% at 0.5M step, and dereased to 88% after 1M step training.
> >
> > > All in all, these additional experiments further showed that diffusion models are capable of constructing strong generalizing representation for underlying factor (e.g. rules), even when only have seen a limited portion of data.
> >
> > thank you for performing these experiments and getting these interesting results.
> >
> > to interpret a bit:
> >
> > 1. there is compositional generalization going on, and it goes beyond leave one t
> > 2. from the name, I assume you did balanced subsampling? did you get any insight on how the model performs if a given factor is only visible 2 or 3 times vs others being much more presnet?
> > 3. the nonlinear learning dynamic could mean that the model finds the "correct" representation early on, but cannot use it for the task of memorizing the dataset?

---

> > ### Comment · Reviewer_Cqnf · 2024-11-25
> >
> > >On the other hand, we think the conditional generation / panel completion experiment can be framed as reasoning, since it potentially involves inferring the underlying rules from observed samples and applying it to the occluded row. (which is also the underlying assumption for the corresponding task in human.) One could argue that the sampling steps in the diffusion or autoregressive model could be analogous to reasoning steps. Further previous works also frame RAVEN's task as abstract visual reasoning [^1][^2], so we think it’s justified to call the task reasoning task.
> >
> > I would only agree with this if there is strong evidence of _abstraction_ as well as multi-step stability. For the review, the non-linear training dynamics make me somewhat comfortable with accepting the argument if phrased more precisely ( e.g. "compositional reasoning" as in your reference [1] ). But in general, I think as we get more and more capable systems, I think it becomes more important to define our terms: what this paper has shown is
> >
> > 1. diffusion models are very good at picking up representations from which classifiers can generalizably learn to classify rules and atributes (compositional modeling)
> > 2. they are able to generate _internally consistent_ rows based of these  (compositionally consistency recall)
> > 3. they struggle with _consistently conditioning_ on a given input
> >
> > In your reference, the authors talk about
> >
> > >attacking visual systems’ major weaknesses in short-term memory and compositional reasoning [22].
> >
> > which, if we break this down, we can call 1+2 compositional reasoning in the sense of learning some fixed rules and 3 the "short term memory" or "grounding". Thus I'd also be somewhat comfortable if the authors choose to frame what is learned as "ungrounded reasoning" or "rule consistent hallucinations" or something along the lines.
> >
> > On all of these, i would find it important to note that this tests the ability on a relatively "simple" task which is "leave one out" generalization. Not a request for this paper, but something for the limitations and future work section, it would be interesting to construct a dataset which has ambiguous completions (due to hints being left out), and see whether the model can learn to generate roughly 50/50 completions (or 1/3 or whatever mulitplicity is constructed). This would further separate the task from "compositional image generation, without the strict need to learn abstract rules since visual similarity might be enough", since the modes of the muliplicity might be visually distinct.

---

> ### Author Response · Authors · 2024-11-18
> **Thanks for the review. Results from New Training Experiments addressing increasing heldout rules, reproducibility across runs and factorized readouts [Part4]**
>
> **Table 3. Sample level consistency and reproducibility of DiT models. S1-3 denotes DiT-S repeat 1-3, B1-3 denotes DiT-B repeats 1-3. We show pairwise comparison of 10240 samples from two models with the same noise seed.**
> entry_match: Fraction of attribute match across all 2488320 attribute entries.
> exact_sample_match: Fraction of samples that match on every attribute, out of 10240 samples.
> sample_C3_match: Fraction of samples with matching C3 rules.
> row_rule_match: Fraction of rows with matching rule set (or invalid), out of 30720 rows.
>
> |  | entry_match | exact_sample_match | sample_C3_match | row_rule_match |
> | --- | --- | --- | --- | --- |
> | S1 vs S2 | 0.765 | 0.00 | 0.805 | 0.706 |
> | S2 vs S3 | 0.767 | 0.00 | 0.817 | 0.719 |
> | S1 vs S3 | 0.767 | 0.00 | 0.808 | 0.713 |
> | B1 vs B2 | 1.000 | 1.00 | 1.000 | 1.000 |
> | B2 vs B3 | 1.000 | 1.00 | 1.000 | 1.000 |
> | B1 vs B3 | 1.000 | 1.00 | 1.000 | 1.000 |
> | S1 vs B1 | 0.708 | 0.00 | 0.750 | 0.633 |
> | S2 vs B2 | 0.698 | 0.00 | 0.756 | 0.635 |
> | S3 vs B3 | 0.701 | 0.00 | 0.754 | 0.633 |
> | Total num | 2488320 | 10240 | 10240 | 30720 |
>
>
> **Questions:**
>
> > can you **factorize the readout with a dimensionality constraint and retain the performance**?
> >
>
> We are not exactly sure about the set up the reviewer is asking for, so we ask for clarification. Do you mean, we should use a 10-way classifier for the relationship and a 4-way classifier for attributes and see if we can get the same performance?
>
> We ran the experiment of using a 10-way linear classifier to classify relations and 4-way classifier to classify attributes and make a factorized prediction of the rule. (which is dimension constrained readout, 14d vs 40d)
> We found that at the last layer, representation can linearly classify this information in a factorized fashion: at block11, full 40-way linear readout had accuracy 99.8%, while attribute x relation factorized readout had accuracy 97.6%. So it can retain most of its performance, though not perfect.
> Further, it seems at deeper layers, the gap between 40-way classification accuracy and factorized classification accuracy is smaller, showing that the representation of rules becomes more factorized throughout the layers.
>
> **Tab. 4 Comparison of Full readout accuracy and factorized readout accuracy.** DiT-S/1 model, 4000 samples per rule, 5 heldout. Representation was recorded at 1M steps, t=25.0 with token averaging. Attribute Acc was 4-way attribute classification accuracy, relation acc was 10-way relation classification accuracy.
>
> |  | Full 40-way Readout Rule Acc | Factorized Readout Rule Acc | Attribute Acc | Relation Acc |
> | --- | --- | --- | --- | --- |
> | blocks0 | 0.274 | 0.114 | 0.388 | 0.292 |
> | blocks2 | 0.622 | 0.220 | 0.478 | 0.428 |
> | blocks5 | 0.953 | 0.723 | 0.930 | 0.764 |
> | blocks8 | 0.996 | 0.961 | 0.997 | 0.963 |
> | blocks11 | 0.998 | 0.976 | 0.999 | 0.977 |
> > How do you perform scoring? manually or an automated evaluation pipeline?
> >
>
> We used an automated pipeline. We have code to evaluate whether each row is consistent with any of the 40 rules, so as to extract the set of rules each row is conforming to, and then find rules that are shared among 2-3 rows (i.e. C2, C3).

---

> > ### Comment · Reviewer_Cqnf · 2024-11-25
> >
> > >We used an automated pipeline. We have code to evaluate whether each row is consistent with any of the 40 rules, so as to extract the set of rules each row is conforming to, and then find rules that are shared among 2-3 rows (i.e. C2, C3).
> >
> > could you go into more detail of this pipeline, or point me to the section of the paper that does this evaluation?
> >
> >  >We are not exactly sure about the set up the reviewer is asking for, so we ask for clarification. Do you mean, we should use a 10-way classifier for the relationship and a 4-way classifier for attributes and see if we can get the same performance?
> >
> > what I mean is that if you enforce two separate bottlenecks (input to 10 dimensions, input to 4 dimensions, possibly some MLP layers before to help with the transformation) and then train a minimal model on these two (either by directly training to predict the factors, like you described, or by constructing an architecture which will be able to use the "latent features" which you bias to represent the 10/4 dimensional real features via the dimensionality constraint), is the model able to deliver on this?
> >
> > if yes, then there is indeed latent disentangling and generalization happening. if not, it is not  a clear negative, but a failure to confirm a prediction

---

### Meta-Review · Area_Chair_U6vB · 2024-12-19

**Metareview:**

The paper studies how generative diffusion models learn hidden rules for reasoning/compositional generalization. In the private discussion, the reviewers argued that they would have leaned in favor of accepting had the authors engaged in the discussion. Overall, the reviewers thought that the experiments in the current version do not match the broad claims of the paper (concrete suggestion: either do more experiments or tune down the title and conclusions). However, I want to note that the reviewers liked the direction of the paper and they encourage a resubmission with their feedback taken into consideration.

**Additional Comments On Reviewer Discussion:**

The authors did not reply to the reviewers to their satisfaction, only replying to some reviewers and not finishing the discussion with the reviewers they engaged with.

---

### Decision · Program_Chairs · 2025-01-22

Reject